# Coordinated cortical thickness alterations across six neurodevelopmental and psychiatric disorders

Neuropsychiatric disorders are increasingly conceptualized as overlapping spectra sharing multi-level neurobiological alterations. However, whether transdiagnostic cortical alterations covary in a biologically meaningful way is currently unknown. Here, we studied co-alteration networks across six neurodevelopmental and psychiatric disorders, reflecting pathological structural covariance. In 12,024 patients and 18,969 controls from the ENIGMA consortium, we observed that co-alteration patterns followed normative connectome organization and were anchored to prefrontal and temporal disease epicenters. Manifold learning revealed frontal-to-temporal and sensory/limbic-to-occipitoparietal transdiagnostic gradients, differentiating shared illness effects on cortical thickness along these axes. The principal gradient aligned with a normative cortical thickness covariance gradient and established a transcriptomic link to cortico-cerebello-thalamic circuits. Moreover, transdiagnostic gradients segregated functional networks involved in basic sensory, attentional/perceptual, and domain-general cognitive processes, and distinguished between regional cytoarchitectonic profiles. Together, our findings indicate that shared illness effects occur in a synchronized fashion and along multiple levels of hierarchical cortical organization.

The conceptualization of neurodevelopmental and psychiatric disorders has undergone several transformations toward overlapping spectra of psychopathology[1,2] associated with underlying polygenicity, neurodevelopmental etiology, and epidemiological comorbidity[1,3,4]. Efforts to empirically understand their dimensional structure has linked the general liability for mental illness to shared risk factors and common alterations in neurodevelopmental processes, predisposing to the clinical conditions ultimately manifested[5–8]. Coordinated multi-level brain alterations across disorders may explain these phenomenological overlaps and common etiology.

Big-data neuroscience initiatives such as the Enhancing Neurolmaging Genetics through Meta-Analysis (ENIGMA) consortium have facilitated large-scale transdiagnostic investigations to identify shared and disorder-specific brain alterations[9]. These studies consistently report cortical thickness alterations in neurodevelopmental and psychiatric disorders[10–15], which serves as a proxy measure for neuronal

density, cytoarchitecture, and intracortical myelination[16–18]. Crucially, previous ENIGMA findings suggest that regional morphological alterations are not only shared between disorders[19–22], but also in part associated with shared genetic etiology[20], regional pyramidal-cell gene expression[21], microstructure, and neurotransmitter system organization[22]. While these findings highlight regional overlaps as shared effects between disorders, the current study aims to address inter-regional dependencies capturing coordinated transdiagnostic patterns of illness effects. That is, differences in brain morphology and function observed in psychiatric patients appear to follow network-like patterns constricted by underlying connectome organization[23–25]. According to the nodal stress hypothesis, highly interconnected regions ('hubs') show increased susceptibility to pathological processes due to shared metabolic alterations, spread of pathogens, or similar gene expression profiles[25,26]. In addition, regional disruptions can act as 'disease epicenters' by promoting pathological processes in

✉e-mail: m.hettwer@fz-juelich.de; s.valk@fz-juelich.de

areas they connect to, thus constituting anchors of network-like alterations[27]. Although the role of network characteristics for cortical alterations in psychopathology is well established[25,28,29], it remains unknown how cross-disorder morphological alterations are embedded in a joint co-alteration network, and whether organizational principles shaping such a network link to underlying neurobiology.

An intuitive approach capturing inter-regional dependencies of illness effects is structural covariance of cortical thickness alterations, which forms cortex-wide co-alteration networks. While structural covariance partly reflects synchronized and genetically coupled maturation during healthy neurodevelopment[30–32], consolidated atrophy in illness also occurs more frequently in regions with high structural covariance[33,34]. Moreover, inter-regional similarities in cortical features tend to be hierarchically organized: Previous mappings of low-dimensional cortex-wide gradients have described continua of cytoarchitectural complexity, long- versus short-distance connectivity, cell density, transcriptomic expression, and phylogenetic and onto-genetic timing[35–38]. Such gradients (or 'axes') compactly summarize covariance patterns via connectome decomposition techniques[35,39], and place brain regions with similar covariance profiles closer together in a common coordinate-frame, regardless of their position on the cortex. These axes offer insights into the global arrangements of cortical features and appear to be distorted in several neuropsychiatric conditions[22,40–42]. Together, previous research highlights the role of convergent hierarchical neurobiological profiles as a central feature of healthy brain organization. Yet, it is currently unknown whether the global arrangement of regional vulnerability to mental illness follows a hierarchical organization as well.

In this study, we identified hubs of transdiagnostic co-alteration networks and disease epicenters using meta-analytical maps for six neurodevelopmental and psychiatric disorders (autism spectrum disorder (ASD), attention-deficit/hyperactivity disorder (ADHD), major depressive disorder (MDD), schizophrenia spectrum disorders (SCZ), bipolar disorder (BD), and obsessive-compulsive disorder (OCD)), provided by the ENIGMA consortium[10–15]. We further employed a cortex-wide gradient mapping approach to identify hierarchical cortical arrangements of transdiagnostic illness effects. Last, we contextualized derived gradients with cytoarchitectonic and functional cortical profiles for multi-level evaluation and studied the embedding of individual disorder impact maps within our framework. We performed multiple robustness checks to evaluate the stability of our findings.

## Results

### Transdiagnostic co-alteration hubs inform disease epicenters

Consistent with previous work[19,21,22], we selected six neurodevelopmental and psychiatric disorders for which illness effects have been studied in large samples in collaborative international meta-analyses by the ENIGMA consortium. To study coordinated trans-diagnostic effects of illness on cortical thickness, we accessed summary statistics from 12,024 patients with ASD[10], ADHD[11], MDD[12], SCZ[13], BD[14], or OCD[15], and 18,969 unaffected individuals from previously published ENIGMA studies (see Table S1). Analyses were restricted to adult samples, except for ASD for which available summary statistics included all age groups. See Table S2 for information on sample demographics. For each condition, we retrieved a Cohen's d map via the ENIGMA Toolbox[43] reflecting case-control differences in cortical thickness for 68 Desikan-Killiany parcels[44] (Fig. 1A). Cohen's d maps were corrected for different combinations of covariates including age, sex, site, and intelligence quotient (Table S2). For contextualization with normative network properties, we further accessed healthy control cortico-cortical structural (diffusion-weighted tractography; DTI) and functional (resting-state functional magnetic resonance imaging; rs-fMRI) connectivity data from an independent sample of healthy young adults from the

Human Connectome Project (HCP[45]) through the ENIGMA Toolbox[43] (Fig. 1B).

First, we computed a transdiagnostic co-alteration matrix by correlating Cohen's d values between regions and across disorders. Regions showing a high sum of strong connections (i.e., correlations) were identified as co-alteration hubs (Fig. 1C). Transdiagnostic hub regions predominated in bilateral medial temporal gyrus and ventral temporal cortex, and more widespread in temporal and frontal regions. When studying which regions are most strongly and consistently affected across disorders via the sum of normalized illness effect maps ('hit map'; see Fig. S1), we observed a significant correlation with transdiagnostic co-alteration hubs ($r = 0.42$, $p_{spin} < 0.0001$), suggesting that hubs are placed in regions with shared impact. This effect predominated for shared thickness reductions ($r = 0.334$; $p_{spin} = 0.01$) rather than relative increases ($r = -0.30$; $p_{spin} = 0.02$). Furthermore, the spatial pattern of co-alteration hubs correlated with normative functional connectivity hubs ($r = 0.50$, $p_{spin} < 0.0001$), but less so with structural hubs ($r = 0.18$, $p_{spin} = 0.08$). Co-alteration hubs were comparable at different thresholds and when correcting for sample size (see Fig. S2).

Having confirmed a general convergence between hubs of coordinated cortical thickness alterations and normative connectome organization, we next investigated whether these patterns are anchored to potential disease epicenters. As previous work has indicated, epicenter mapping aids to understand how the normative connectivity profile associated with a specific region may play a central role in the manifestation of a disorder[27,46,47]. Here, we identified transdiagnostic epicenters as regions whose connectivity profile may underlie illness effects that are consistently organized across disorders, i.e., regions whose network embedding correlates significantly with co-alteration hubs. Thus, the epicenter mapping approach highlights the role of regions that do not necessarily constitute hubs themselves[48] but may contribute to shaping shared patterns of illness effects through strong or distributed connections with co-alteration hubs. Systematically investigating connectivity profiles of 68 cortical seeds revealed primarily temporal and prefrontal regions as potential transdiagnostic disease epicenters (Fig. 1D). This finding held true when computing epicenters based on the 'hit map' (Fig. S1E, F). Highest ranked functional disease epicenters were observed in the left entorhinal cortex, left pars orbitalis, right banks of the superior temporal sulcus (STS), left pars triangularis, and left STS ($r = 0.55-0.59$; all $p_{spin} < 0.05$). Top five structural disease epicenters were present in left pars opercularis and triangularis, inferior parietal lobe, STS bank, and caudal middle frontal gyrus ($r = 0.28-0.42$; all $p_{spin} < 0.05$).

### Macroscale gradients of transdiagnostic co-alteration networks

So far, our analyses suggest that the cortex-wide network of trans-diagnostic illness effects is non-randomly organized, with hubs of prominent covariance and epicenters shaping the co-alteration network. Next, via manifold learning, we sought to study the embedding of these features within low-dimensional organizational gradients[39,49]. This analysis was based on the same co-alteration matrix used to derive transdiagnostic hubs (Figs. 1C and 2A). We applied diffusion map embedding[49] to project regional and long-range connections within covariance networks into a common space. This yielded unitless components, each of which denotes the position of nodes on a continuum describing similarities in regions' structural covariance profiles (Fig. 2A). Thus, opposing apices of a gradient reflect maximally divergent covariance patterns.

The principal gradient of transdiagnostic covariance (G1) captured a dominant dissociation between frontal and temporal lobes and accounted for 36% of variance in transdiagnostic co-alteration (Fig. 2B). The secondary gradient (G2) spanned from occipito-parietal regions to temporo-limbic structures, explaining 21% of variance. Findings were comparable at different thresholds and robust

**A** Cortical thickness alterations

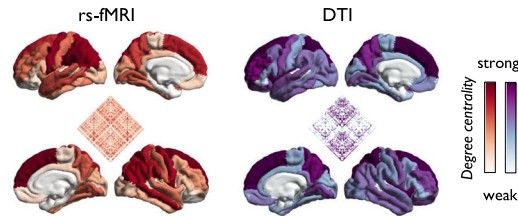

**B** HCP connectivity data

**C** Co-alteration network hubs and epicenter mapping

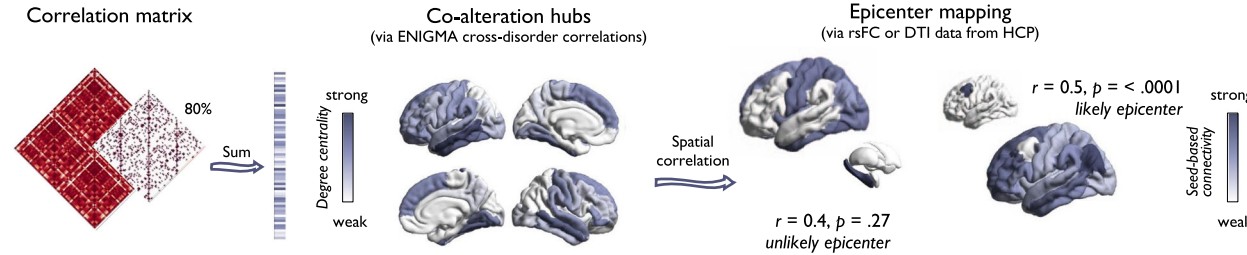

**D** Transdiagnostic disease epicenters

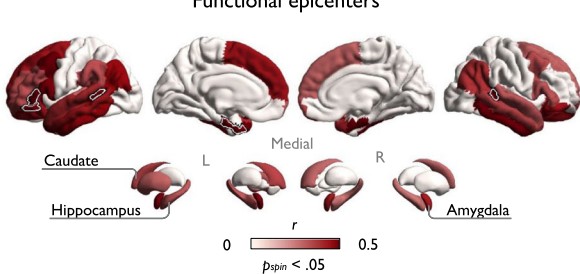

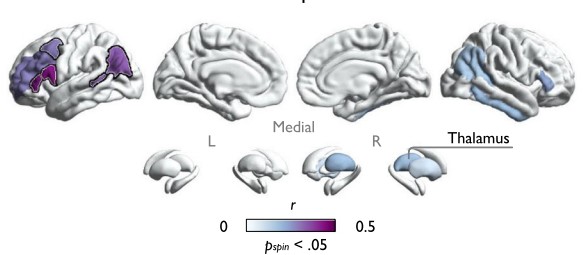

**Fig. 1 | Hubs and epicenters shaping transdiagnostic co-alteration patterns.**
**A** Disorder-specific Cohen's d maps indicating case-control differences in cortical thickness. **B** Normative connectivity matrices derived from resting-state functional magnetic resonance imaging (rs-fMRI) and diffusion-weighted tensor imaging (DTI) from the Human Connectome Project (HCP)[45] and hubs (degree centrality). **C** Left: Computation of co-alteration hubs. Degree centrality was computed as the sum of above-threshold (80%) connections at each parcel using disorder maps from the Enhancing Neuroimaging Genetics through Meta-analyses (ENIGMA) consortium. Right: Visualization of the epicenter mapping approach using resting state functional connectivity (rsFC) or DTI. Seed-based connectivity profiles were systematically correlated with co-alteration hubs (using Pearson's r and assessing

significance via two-sided spin-tests, correcting for spatial auto-correlation, without further correction for multiple comparisons). **D** Transdiagnostic disease epicenters are depicted as correlations between co-alteration hubs and HCP normative seed-based connectivity profiles (rs-fMRI or diffusion tensor imaging (DTI)), thresholded at $p_{spin} < 0.05$ (this panel shows DTI examples). High correlations imply high likelihood of a structure constituting a disease epicenter. Top five functional and structural disease epicenters are framed in white/black. Source data are provided as a Source Data file. ADHD = Attention-deficit/hyperactivity disorder; ASD = Autism spectrum disorder, BD = Bipolar disorder, MDD = Major depressive disorder, OCD = Obsessive-compulsive disorder, SCZ = Schizophrenia spectrum disorders.

against parameter manipulation, sample size correction, and selection of diagnoses (see Fig. S2). An overview of all computed gradients is presented in Fig. S3. Investigating the correspondence between the disease epicenters and the transdiagnostic gradients, we found that the apices of G1 captured previously identified functional disease epicenters (Fig. S4). This implies that frontal and temporal epicenters each contribute to the overall pattern of co-alterations but do so in a maximally distinct manner (Fig. S5).

Since previous studies have shown that cortical thickness alterations in psychopathology are more prominent in regions with high structural covariance[50], we assessed whether the disease-related relative changes in cortical thickness align with normative organization of absolute cortical thickness. Indeed, we observed a correlation between the principal cortical thickness covariance gradient (anterior-posterior; Fig. 2C)[36] and G1 ($r = -0.74$, $p_{spin} = 0.0015$) but not G2 ($r = -0.11$, $p_{spin} = 0.27$). The second cortical thickness covariance gradient (inferior-superior) was not related to G1 ($r = 0.32$, $p_{spin} = 0.21$) or G2 ($r = -0.25$, $p_{spin} = 0.07$).

## Microstructural and transcriptomic contextualization

After capturing macroscale organization of disease effects, we contextualized identified gradients with microscale cytoarchitecture to gain a multi-level understanding of neurobiological cortical profiles in shaping transdiagnostic co-alteration networks. To this end, we stratified our gradients according to von Economo-Koskinas cytoarchitectonic classes[51]. We observed a prominent distinction between granular and agranular cortices across our principal transdiagnostic gradient (Fig. 2D), whereas G2 distinguished between granular and parietal cytoarchitectonic classes.

Using post-mortem gene expression data from the Allen Human Brain Atlas (AHBA) as a reference[52], we next identified genes for which spatial expression patterns significantly correlated with G1 (see Table S3). This approach has previously revealed genetic links to normative brain development and organization[52–54] as well as structural abnormalities in disease[42,55,56]. We generated null models to assess spatial specificity (including spatially autocorrelated phenotype maps[57]) and gene specificity (including (i) genes with similar levels of

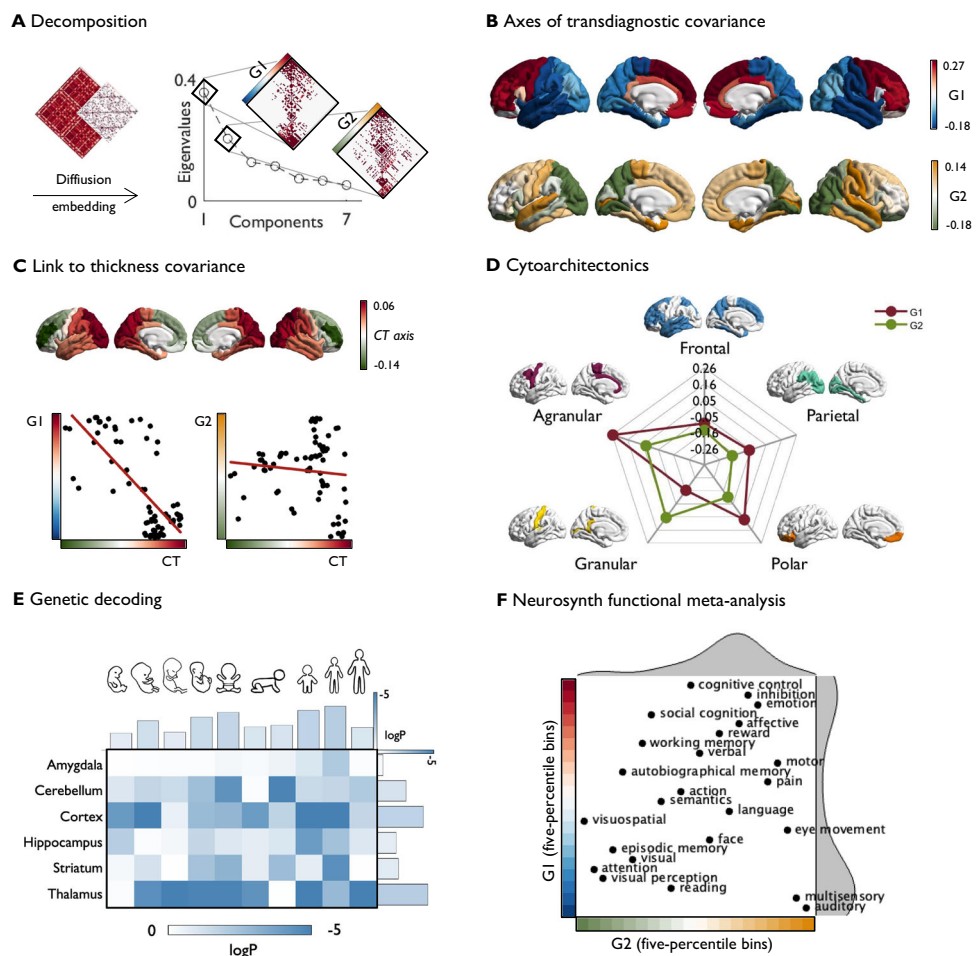

**Fig. 2 | Macroscale organization of transdiagnostic covariance in cortical thickness alterations. A** A cross-disorder structural covariance matrix was thresholded at 80% and decomposed using diffusion map embedding. Covariance along the principal (G1) and second (G2) gradients is depicted on the right. **B** Transdiagnostic gradients G1 and G2. **C** Correlation between a normative axis of cortical thickness (CT) covariance[36] and transdiagnostic gradients. **D** Cross-condition gradients stratified according to von Economo-Koskinas cytoarchitectonic classes[51]. **E** Developmental gene enrichment analysis based on 232 genes for which spatial expression patterns correlated with G1 (of which 146 showed a positive correlation, i.e., were overexpressed in prefrontal compared to temporal regions). **F** Meta-analysis for diverse cognitive functions obtained from NeuroSynth[60]. We computed parcel-wise z-statistics, capturing node-function associations, and calculated the center of gravity of each function along 20 five-percentile bins of G1 and G2. Function terms are ordered by the weighted mean of their location along the gradients. Source data are provided as a Source Data file.

coexpression and (ii) genes overexpressed in the brain compared to the rest of the body)[58] of the identified gene set. Out of 232 genes for which expression patterns correlated significantly with G1, 146 showed a positive correlation with G1, i.e., they were more strongly expressed in the PFC than in temporal regions. Developmental gene enrichment analysis[59] revealed that next to the cortex, identified genes were most prominently expressed in the thalamus and cerebellum across various developmental windows (Fig. 2E). In a combined assessment of all brain structures, genes appeared to be enriched most strongly during neonatal early infancy, mid/late childhood, and adolescence. G2 was not significantly associated with genes included in the AHBA after correcting for spatial and gene specificity.

## Associations with task-based functional activations

Next, we aimed to identify potential functional implications by investigating whether transdiagnostic gradients dissociate regions associated with distinct functional engagement. To this end, we conducted a meta-analysis on task-specific functional activations for 24 cognitive terms using the NeuroSynth database[60]. We binned each gradient into five-percentile bins and defined regions of the same bin as a region of interest (ROI). Resulting 20 ROIs for each gradient were then tested for their overlap with meta-analytic ROIs associated with each of the 24

cognitive terms via z-statistics. The magnitude of an average z-value at a ROI (i.e., a position along the gradient) reflects the strength of its association with a certain functional task activation. We sorted the topic terms by their weighted mean position along both gradients, revealing systematic shifts in functional networks along transdiagnostic axes of co-alteration. In a combined 2D space framed by both gradients, we could distinguish between different co-alteration patterns in primary (e.g., 'auditory') and 'multisensory' regions at the temporal apex, higher-order perceptual structures (e.g., 'visual perception' and 'attention') at the occipito-parietal apex, and complex cognitive functions (e.g., 'cognitive control' and 'inhibition') at the frontal apex (Fig. 2F and Fig. S6).

## Embedding of individual disorders within a transdiagnostic co-alteration space

Having established several features guiding a transdiagnostic co-alteration network, we last aimed to evaluate the positioning of individual disorders within this continuous transdiagnostic space. To this end, we first studied the correspondence between a parcel's whole-brain transdiagnostic covariance profile and a parcel's whole-brain disorder-specific covariance profile (see Fig. 3A, B). While associations with the transdiagnostic pattern vary between

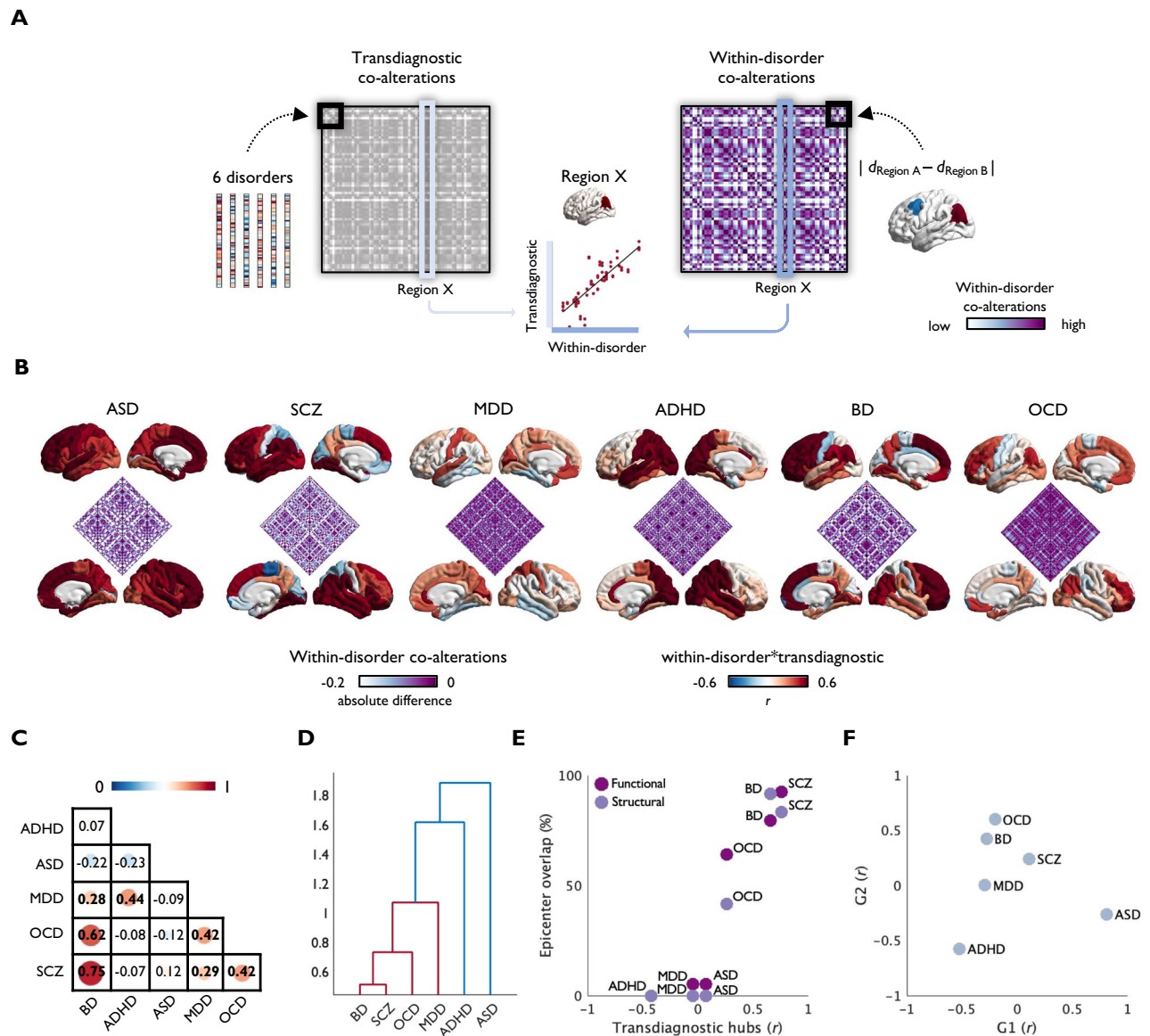

**Fig. 3 | Embedding of six disorders within transdiagnostic co-alteration networks. A** Computation of transdiagnostic and within-disorder co-alteration matrices. **B** Region-wise correspondence between disorder-specific and transdiagnostic co-alteration profiles. Disorder-specific inter-regional difference scores were inverted so that higher correlations with transdiagnostic patterns indicate higher coupling. **C** Similarity of illness effects between disorders, i.e., correlations of Cohen's d maps, and how they cluster together in a two-cluster solution (**D**). Position of individual disorders within a transdiagnostic co-alteration space based

on **E** the correlation between transdiagnostic hubs and Cohen's d maps (x-axis) and the overlap between transdiagnostic and disorder-specific epicenters (y-axis); and **F** the correlation between the principal (G1) and secondary (G2) transdiagnostic gradients with Cohen's d maps. Source data are provided as a Source Data file. ASD = Autism spectrum disorder, SCZ = Schizophrenia spectrum diagnoses, MDD = Major depressive disorder, ADHD = Attention-deficit/hyperactivity disorder, BD = Bipolar disorder, OCD = Obsessive-compulsive disorder.

disorders and across the cortex, most disorders showed highest similarity to shared patterns in heteromodal cortices. This mirrors other findings presented here which suggest heteromodal cortices as regions that not only tend to be affected, but also tend to be affected similarly across disorders and in a synchronized manner across the cortex. Next, we compared the degree of similarity between disorders and their embedding within the transdiagnostic co-alteration space. Replicating what previous transdiagnostic studies have shown[19,20], we observed a cluster composed of SCZ, BD, and OCD, while ADHD and ASD stayed separate (Fig. 3C, D). In contrast to clustering approaches, our cross-disorder covariance approach aimed to describe a transdiagnostic organizational space in which disorder effects occur. Indeed, we found that disorders that cluster together, such as SCZ, BD, and OCD showed a similar placement within this transdiagnostic co-alteration framework (see

Fig. 3D–F). While transdiagnostic hubs correlated with illness effect maps in SCZ ($r = 0.76$, $p_{spin} < 0.0001$), BD ($r = 0.66$, $p_{spin} = 0.001$), and OCD ($r = 0.26$, $p_{spin} = 0.03$), this was not the case for ASD ($r = 0.07$, $p_{spin} = 0.31$) and MDD ($r = -0.05$, $p_{spin} = 0.41$), and ADHD showed a negative correlation ($r = -0.42$, $p_{spin} = 0.003$). Similarly, disorder-specific epicenters overlapped with transdiagnostic epicenters in SCZ, BD, and OCD, and in part in MDD (see Fig. 3E and Fig. S7), whereas ADHD and ASD showed no significant disorder-specific epicenters in the first place. Together, these analyses indicate that similar illness effect maps relate to similar degrees to transdiagnostic co-alteration hubs, are linked to epicenters that overlap to similar degrees with transdiagnostic epicenters (Fig. 3E), and are positioned more closely together in a transdiagnostic covariance space framed by G1 and G2 (Fig. 3F and Table S4). However, we also observe that disorders which show some similarity but are

allocated to different clusters, such as MDD and ADHD, are positioned closer to each other in our continuous transdiagnostic space, crossing cluster boundaries.

## Discussion

Our study reports coordinated effects of six major mental disorders (SCZ, BD, OCD, ASD, ADHD, and MDD) on cortical thickness and their association with functionally relevant neurobiological patterns across multiple scales of analysis. Thus, we extended previous investigations of shared regional effects[19–22] toward a network-based approach that embeds regional alterations within cortical hierarchies of transdiagnostic covariance of illness effects.

We identified hubs of transdiagnostic co-alteration predominantly in lateral and ventral temporal lobes, with some impact on post-central and medial frontal regions. Importantly, these hubs overlapped with regions showing shared thickness alterations, indicating especially pronounced coordination within co-alteration networks between consistently affected regions. Observing an interrelationship of pathological cortical thickness alterations between temporal and prefrontal heteromodal cortices, but less so in unimodal and paralimbic cortices, indicates distinguishable processes shared between disorders and across the cortex. Furthermore, co-alteration hubs followed the spatial pattern of normative functional connectivity hubs, suggesting that captured variability in susceptibility may link to nodal stress[25,26,61]. Indeed, in vivo markers of e.g., aberrant energy metabolism and post-mortem proteomic analyses have revealed overlaps between MDD, SCZ, and BD[62–64]. As hubs are more strongly influenced by genes than non-hubs[65], hub regions may exhibit increased shared vulnerability for atypical neurodevelopment, supported by both the polygenicity and genetic overlaps in psychiatric diagnoses. Thus, nodal stress, along with other potential factors such as shared genetic susceptibility, appears to be a strong candidate explanation for the irregular topographic distribution of covarying illness effects[28,66].

Present results further indicate that large-scale patterns of shared illness effects are shaped by both structural and functional epicenters. That is, transdiagnostic epicenters suggest a central role of prefrontal and temporal cortex in the manifestation of mental illness, indicating how transdiagnostic cortical alterations are anchored in the connectivity of identified regions. Notably, influences of functional epicenters emerged above and beyond hard-wired tracts. Such a divergence is likely[67], as functional connectivity reflects a temporal correlation of activity which may be driven by distant input into a spatially distributed polysynaptic network[68,69]. The high concordance of prefrontal and temporal connectivity profiles with co-alteration hubs indicates that epicenters preferentially emerged in regions known to extend long-range connections[70], facilitating their contribution to cortex-wide organizational patterns. Structures in the mediotemporal lobe and ventrolateral PFC were identified as most likely epicenters. Both regions have been implicated in cognitive impairments and developmental susceptibility across neurodevelopmental and psychiatric disorders[53,71,72]. Mediotemporal structures further act as nodal points between multimodal cortical association areas and the subcortex, and feature transitions in cytoarchitecture from iso- to allocortical regions[37]. These features may increase both vulnerability to nodal stress and the spread of pathological alterations through wide-ranging connections. At the same time, the vlPFC shows protracted plasticity throughout multiple neurodevelopmental stages[73]. While allowing for continuous refinement of complex cognitive abilities, protracted plasticity gives room for aberrant maturational processes leaving the individual more susceptible to developmental aberrations. Overall, the epicenter mapping approach thus identified anchors of large-scale transdiagnostic co-alteration networks in regions that both have the potential to spread illness effects through long-range connections and are susceptible to maturational aberration.

Further investigating transdiagnostic covariance via manifold learning, we recapitulated cortex-wide gradients along which co-alteration patterns were organized. The principal transdiagnostic gradient captured a cortex-wide segregation of frontal and temporal structures, indicating that cortical thickness alterations in both regions are embedded in maximally distinct covariance networks. The concordance of G1 with the normative organizational axis of cortical thickness covariance[36] mirrors previous findings indicating increased susceptibility to cortical atrophy in regions that exert high structural covariance[33,74]. As cortical thickness covariance is assumed to reflect common maturational trajectories[30], atypical neurodevelopment likely contributes to shaping cortical gradients of co-alteration networks. The process of shaping transdiagnostic gradients throughout development may further be influenced by subcortico-cortical circuits[75–78], as suggested by our transcriptomic decoding findings. Here, we observed that genes whose expression pattern aligns with G1 are also enriched in the cerebellum and thalamus in early developmental phases. Notably, studies on subcortical interactions have linked impaired functional coordination within cerebello-thalamo-cortical circuits to a general liability for psychopathology[79,80]. It is thus possible that the organization of transdiagnostic co-alterations observed in the cortex partly builds upon alterations in subcortical circuits. The secondary gradient was restricted to uni- and heteromodal sensory cortices in the posterior cortex, segregating regions that hold primary sensory (pericalcarine cortex, post-central and superior temporal gyrus) and paralimbic (entorhinal) cortices from multimodal association regions in the occipito-parietal cortex. Both axes described segregations along different cytoarchitectural classes. Whereas G1 traversed between agranular, paralimbic, versus granular, primary cortices, G2 showed a cytoarchitectural divergence between granular and frontal/parietal cortices. Variable susceptibility to disease impact thus suggests that areas with shared cytoarchitecture are more likely embedded similarly within pathological networks. This may be due to similar local computational strategies supported by cell count and wiring strategies[17], development[81], and the degree of plasticity associated with different degrees of cortical lamination[53]. Future work may further investigate the specific neurobiological mechanism linking cytoarchitecture, function, and mental illness.

We further contextualized our findings with respect to functional processes through meta-analytical task-based activations. Combining G1 and G2 in a two-dimensional space revealed distinct co-alteration profiles at three levels of information processing, i.e., primary/multi-sensory, perception/attention, and domain-general cognitive control. Interestingly, all three levels show various processing impairments in different neuropsychiatric conditions which are in part interrelated: Firstly; atypical early development of sensory cortices can contribute to social cognitive deficits through impaired social cue perception[82,83] and, more generally, deficits in multisensory binding[83,84]. Secondly; at an intermediary level, aberrant functional involvement and structural integrity of attention networks have been identified as a prominent transdiagnostic feature of neuropsychiatric conditions[28,85]. Thirdly; upstream consequences of dysregulated attention networks ultimately contribute to impaired higher-order cognitive functions such as executive control. Impaired executive control does not only constitute a transdiagnostic feature in mental illness[86], but is also a predictor of cognitive and socio-occupational impairment[86–88]. Despite inter-related deficits within functional networks, the observation that multiple processing levels are associated with distinct structural co-alteration patterns indicates independent maturational causes and distinct vulnerability. In line with findings from cytoarchitectonic contextualization, levels of functional engagement of cortices involved in similar tasks appear to leave brain regions processing similar types of information with shared susceptibility. Given that

sensory regions develop earlier than association regions in the cortical maturational sequence[89], differences in pathological covariance profiles may link to the degree to which their developmental peaks overlap with vulnerable periods for neurodevelopmental and psychiatric disorders. This raises the question whether identified cortical gradients also reflect a spatiotemporal gradient of atypical neurodevelopment and inspire respective investigations in longitudinal/prospective studies. Overall, our findings indicate that the degree to which regional alterations may be linked to and potentially facilitate alterations in other brain regions (i.e., potential epicenters), and the degree to which such interrelations pose a general feature of the neurodevelopmental and psychiatric disorders included (i.e., co-alteration hubs) appears to vary across the cortex and follows general neurobiological principles of brain organization (i.e., cortex-wide axes).

Last, we aimed to investigate how the proposed transdiagnostic co-alteration space, framed by both transdiagnostic covariance gradients, compares to previous descriptions of cross-disorder similarities and disorder clusters. That is, our cross-disorder covariance approach generates a continuous space within which disorders vary with respect to their topography of similarity to transdiagnostic patterns across the cortex. While we indeed found that positions of disorders within this space converge with their allocation to disorder clusters, the co-alteration framework captures both similarities within and between clusters in a continuous manner. By embedding illness effects within a space shaped by genetic and maturational processes, we gain further insight in differentiable neurobiological mechanisms underlying individual disorders. Indeed, the first gradient, stretching between frontal and temporal regions showed similarities with a previously described anterior-posterior axis along the cortical mantle[90]. Previous work has indicated differentiable spatial patterns of co-maturation and development along several spatial axes, indicating the interplay of multiple neurodevelopmental mechanisms across the cortex[56,90,91]. The observed systematic alterations along such axes across disorders may reflect differential disruptions in pre- and postnatal neurodevelopment. Moreover, we observed that, for most disorders, overlap between disorder-specific and transdiagnostic covariance is highest in heteromodal cortices. This convergence may be linked to their placement within these neurodevelopmental axes[53,92], supporting these regions as targets of transdiagnostic investigations. Future work may evaluate potential causes and critical time windows of development within this framework, enhancing our understanding of the ontogeny of cortical organization in health and disorder.

It is of note that, although disorder impact generally converged in heteromodal regions and linked to transdiagnostic covariance gradients, each disorder showed a unique embedding within our framework. For example, though we observed widespread coupling between transdiagnostic and disorder specific covariance networks in ASD and ADHD, and marked association with the principal transdiagnostic covariance gradient, there was only reduced correspondence with the epicenter framework, indicating disrupted relationship between disorder hubs and connectivity profiles. Conversely, MDD showed in particular correspondence with transdiagnostic patterns in ventral PFC, subgenual anterior cingulate, somatosensory cortex and nucleus accumbens, but showed reduced correspondence with transdiagnostic epicenters and the transdiagnostic gradients. It is possible that MDD, being at the center of the 2D gradient space and showing highest similarities with both ADHD and OCD, can be best described by yet another axis not captured in the current framework which is dominated by neurodevelopmental patterning. The future work expanding our framework to more disorders as well as atlasses with higher granularity may be able to further pin-point differential axes of embedding for different disorders.

While our findings underline the relevance of transdiagnostic approaches, they do not contradict the existence of etiological and phenomenological differences within and between psychiatric diagnoses. Our transdiagnostic approach does not capture heterogeneity within and between highly related diagnostic categories, as expected to be present e.g., within the included SCZ (schizophrenia spectrum disorders) and BD (type I and II combined) samples. However, shared features crossing diagnostic boundaries are likely also an important factor contributing to within-disorder heterogeneity. Moreover, individuals may be diagnosed with multiple different disorders across their lifespan[4]. Understanding which neurobiological principles drive the spectrum of varying neurodevelopmental and psychiatric disorders is a crucial piece of the puzzle of the biological origin of disorder variability. Yet, investigating both disorder-specific phenomena and heterogeneity within (spectrum-)diagnoses forms a crucial line of research that will continue to complement our transdiagnostic findings. Presented cortex-wide co-alteration features shall facilitate and provide a transdiagnostic coordinate frame for such insights.

Although we mostly included adult samples and age-corrected summary statistics, there are some offsets among mean ages of ENIGMA maps and between ENIGMA maps and the reference data from other sources (e.g., HCP). These offsets potentially influenced parameters known to change during development and aging, such as hub organization[89]. In addition, neurodevelopmental conditions have different mean ages of onset so that patients included have certainly experienced different lengths of disease and medication histories. It should further be noted that also other disorders such as substance abuse or anxiety disorders tend to co-occur with some of the disorders included here, but could not be included in the analyses as ENIGMA cortical thickness summary statistics have not yet been published. Further work including a wider range of disorders will help to evaluate the generalizability of our transdiagnostic model. ENIGMA summary statistics used here are based on the Desikan-Killiany atlas[44]. They thus contain comparatively sparse data points across the cortex and summarize data from broader areas that contain a mosaic of neurobiological regions that may be differentially affected by disease. Moreover, differences in parcel size[93], measurement error, subject motion and scanner/site effects[94,95] may slightly influence spatial covariance analyses. Last, the lack of subject-level clinical information impeded the direct assessment of clinical implications of current findings. However, understanding the principles according to which cortical alteration patterns are organized across diagnoses will provide a fruitful basis for further investigations on the interrelationship between network organization and symptoms shared across disorders, as well as variations within categorical diagnoses.

In sum, our findings highlight the value of linking multiple neurobiological levels of information—from macroscale neuroimaging to microscale transcriptomic data—to identify systematic transdiagnostic patterns of illness effects. Investigating these patterns revealed coordinated cortical alterations across conditions that are shaped by connectomic, cytoarchitectonic, and functional characteristics. As such, we provide a cortical coordinate system in line with concepts of dimensional psychiatry and network-based pathology, to which future clinical neuroscience findings can be aligned and integrated. Future work may further expand on this approach not only to include different modalities and neuroimaging metrics (e.g., surface area and subcortical structures), but also to consider a much wider range of conditions and age ranges, which is now becoming increasingly possible due to the availability of multi-disease consortia and datasets[96,97]. This may provide a crucial step toward understanding the neuroetiology of neuropsychiatric conditions.

## Methods
### ENIGMA neuroimaging summary statistics
For our transdiagnostic analyses, we used publicly available multi-site summary statistics published by the ENIGMA Consortium, and available within the ENIGMA Toolbox (https://github.com/MICA-MNI/

ENIGMA[43]). Included neurodevelopmental and psychiatric disorders comprised ADHD[11] ($n_{cases}$ = 733, $n_{controls}$ = 539), ASD[10] ($n_{cases}$ = 1571, $n_{controls}$ = 1651), BD (type I and II, cumulated)[14] ($n_{cases}$ = 1837, $n_{controls}$ = 2582), MDD[12] ($n_{cases}$ = 1911, $n_{controls}$ = 7663), OCD[15] ($n_{cases}$ = 1498, $n_{controls}$ = 1436), and SCZ (including schizophrenia spectrum diagnoses)[13] ($n_{cases}$ = 4474, $n_{controls}$ = 5098). Except for ASD for which available summary statistics included all age groups, we restricted our analyses to adult samples. This decision may increase the variance in disease duration due to differences in typical ages of onset associated with the six diagnoses. However, we aimed to match adults to minimize effects that are linked to development and aging, which are potentially larger than the effects of disease duration. We based our analyses on covariate-adjusted case-control differences denoted by across-site random-effects meta-analyses of Cohen's d-values for cortical thickness. Age, sex, and site information was fitted to cortical thickness measures via multiple linear regression analyses. As previous studies have shown associations between IQ and brain structure as well as alterations of this association in ASD[98], IQ was included as a covariate in the ASD sample. See Table S2 for an overview on demographics and study-specific covariates. Preceding the computation of summary statistics, raw data was pre-processed, segmented and parcellated according to the Desikan-Killiany atlas[44] in FreeSurfer (http://surfer.nmr.mgh.harvard.edu) at each site and according to standard ENIGMA quality control protocols (see http://enigma.ini.usc.edu/protocols/imaging-protocols). Sample sizes ranged from 1272 (ADHD) to 9572 (SCZ). We redirect the reader to the original publications[10–15] for more details on age matching and controlling for medication or comorbidities. Ethics approval and subjects' informed consent was obtained by individual cohort investigators.

## Population connectivity data
Functional and structural connectivity matrices were based on 1 h of rs-fMRI and diffusion MRI from healthy adults ($n$ = 207, 83 males, mean age = 28.73 ± 3.73 years), respectively. The data were acquired through the HCP[45], minimally pre-processed according to HCP guidelines[99] and made publicly available as group-average structural and functional connectivity matrices in the ENIGMA Toolbox[43]. See Supplementary Material for more information about the computation of connectivity matrices.

## Structural covariance of disease effects on local brain structure
We derived a 68 × 68 cross-condition correlation matrix by computing inter-regional Pearson's correlations of cortical thickness Cohen's $d$ values across the six conditions.

## Covariance hubs and transdiagnostic disease epicenters
In order to derive co-alteration network hubs using a degree centrality approach, we first identified which connections (i.e., correlations) of the previously derived cross-condition correlation matrix belong to the top 20% of strong connections. We then computed the sum of these connections for each parcel, where regions with many strong connections represent hubs of high transdiagnostic covariance of illness effects (Fig. 2A). Next, we accessed whole-brain functional (rs-fMRI) and structural (DTI) connectivity matrices from a healthy adult HCP dataset[45] via the ENIGMA Toolbox[43], which we also thresholded at 80%. Normative connectivity hub maps based on HCP data was computed using the same degree centrality approach (i.e., the sum of all strong connections) and spatially correlated with the transdiagnostic structural co-alteration hub map. Significance was assessed via spin tests (see Supplementary Material and ref. [57]). This analysis aimed to assess whether co-alteration hub regions align with the normative underlying connectome and may thus be linked to nodal stress.

To identify transdiagnostic disease epicenters, we systematically assessed spatial similarity of each parcel's normative whole-brain connectivity profile with our map of co-alteration hubs using spatial permutation tests. To do so, we collected seed-based functional (rs-fMRI) and structural (DTI) connectivity matrices for each parcel and 14 subcortical structures from the same HCP dataset[45]. We then spatially correlated each structure's connectivity profile with the co-alteration hub map. The higher the spatial similarity between an epicenter's connectivity profile and co-alteration hubs, the more likely this structure represents a disease epicenter (at $p < 0.05$ after spin tests). Resulting likelihoods were ranked to identify the top five structural and functional transdiagnostic disease epicenters.

## Gradient decomposition
We computed macroscale organizational gradients using BrainSpace[49] (https://github.com/MICA-MNI/BrainSpace) in Matlab 2020b. The 68 × 68 structural covariance matrix was thresholded at 80% and transformed into a non-negative square symmetric affinity matrix by using a normalized angle similarity kernel. We then applied diffusion mapping as a nonlinear dimensionality reduction method[39,49] to estimate the low-dimensional embedding of our previously derived high-dimensional affinity matrix. Here, cortical nodes that are close together reflect nodes that are inter-connected by either many supra-threshold or few very strong edges, whereas nodes that are farther apart reflect little or no covariance. We set α, a parameter which controls the impact of sampling density (where 0–1 = maximal to no influence), to 0.5. This α value retains global relations in the low-dimensional space and is assumed to be comparatively robust to noise in the input matrix. Lastly, we assessed the amount of information explained by received gradients, selected the first two gradients for further analyses and projected them onto a cortical mesh using BrainStat (https://github.com/MICA-MNI/BrainStat).

## Link to normative axes of cortical thickness organization
An association with normative cortical thickness organization was studied by correlating derived transdiagnostic gradients with previously established gradients of cortical thickness covariance in healthy adults. These two normative gradients were based on cortical thickness data from individuals in the S1200 HCP sample and were derived using the same diffusion embedding approach as described above (see ref. [36]). Spatial associations were evaluated using spin tests[57].

## Cytoarchitectonic contextualization
To determine whether transdiagnostic gradients recapitulate cytoarchitectonic variation evidenced by post-mortem histological assessments, we further stratified our gradients according to the five von Economo-Koskinas cytoarchitectonic classes[51]. This atlas subdivides the cortex into five categories: agranular (thick cortex housing large cells but scarce layers II and IV), frontal (thick cortex, large but sparse cells, layers II and IV are present), parietal (thick cortex that is rich in cells, dense layers II and IV, slender pyramidal cells), polar (thin cortex, rich in cells, particularly granular cells) and granular/koni-cortex (very thin cortex with highest density of small cells).

## Genetic decoding
Having established macro- and microscale contextualization of our findings, we finally aimed to understand its association with gene transcriptomic data provided by the Allen Institute for Brain Science (AIBS)[52]. Microarray expression data was processed in abagen[100], including intensity-based filtering, normalization and aggregation within Desikan-Killiany parcels and across donors. Only genes with a similarity of $r > 0.2$ across donors were included, resulting in 12,668 genes for the analysis[43]. We correlated trans-diagnostic gradients with the post-mortem gene expression maps and tested for spatial and gene specificity using several null models: First, we generated a set of random spatially autocorrelated phenotype maps[57] to test the spatial specificity of associations observed

between gene transcriptomic profiles and transdiagnostic gradients. Genes with an expression profile significantly correlated with G1 or G2 ($p_{spin} < 0.01$) were defined as gene set for following gene specificity tests. Next, using the Gene Annotation using Macroscale Brain-imaging Association (GAMBA) Toolbox[58], we tested this gene set against two types of null models: The null-coexpressed-gene model and the null-brain-gene model. The null-coexpressed-gene model includes genes with a similar co-expression level as the gene set of interest to generate null distributions. The null-brain-gene model generates null models exclusively from genes over-expressed in brain tissue and is thus more conservative than classical random-gene models. If a gene set was identified as significantly associated with a transdiagnostic gradient in both linear regressions and described permutation tests, it was next used as input for a developmental enrichment analysis via the cell-type specific expression analysis (CSEA) developmental expression tool (http://genetics.wustl.edu/jdlab/csea-tool-2)[59]. This allowed us to compare genes identified with respect to the AIBS repository with developmental expression profiles from the BrainSpan dataset (http://www.brainspan.org), yielding more detailed, yet indirect, information about brain structures and developmental windows in which identified genes are enriched.

## Functional decoding

To assess whether transdiagnostic gradients capture differential impact on cognitive networks, we assessed the distribution of various cognitive functions along transdiagnostic gradients[35,53]. To this end, we conducted a meta-analysis using the NeuroSynth[60] database. Briefly, we derived 20 ROI maps by decomposing G1 and G2 into five-percentile bins and combining regions of the same bin to a joint ROI. The granularity of five-percentile bins is assumed to capture subtle variations along cortical axes. We then examined the association of each ROI with 24 cognitive topic terms via z-statistics. Topic terms were then sorted based on their center of gravity and arranged in a two-dimensional space that was created by merging G1 and G2, for visualization.

## Association between disorder-specific illness effect patterns with transdiagnostic findings

Last, we aimed to understand the degree to which cortical alterations observed in individual disorders are reflected in described transdiagnostic features. To this end, we first examined cross-cortical similarities of illness effects within disorders[101,102], via absolute differences in Cohen's d values between regions. We then correlated each parcel's disorder-specific whole-brain covariance profile with the previously described transdiagnostic covariance profile of the same parcel. This allowed us to investigate disorder-specific cortical topographies of varying regional associations with transdiagnostic patterns. Second, we examined the similarity of illness effect maps among disorders via pair-wise correlations and applied hierarchical clustering to the resulting cross-disorder correlation matrix. These steps allowed us to investigate how disorders with varying similarity to each other and to transdiagnostic features described in this study are positioned in the proposed transdiagnostic covariance space. To this end, we correlated the transdiagnostic co-alteration hub map with disorder-specific Cohen's d maps, and computed disorder-specific epicenters by systematically correlating each region's normative connectivity profile (rs-fMRI and DTI) to disorder-specific Cohen's d maps. We then assessed the overlap between disorder-specific and transdiagnostic epicenters in percent, and combined this with the association to transdiagnostic hubs in a 2D space. Similarly, we examined the correlation between transdiagnostic gradients and disorder-specific Cohen's d maps in a 2D space framed by G1 and G2. Together, these analyses revealed how individual disorders are embedded in relation to each other within a transdiagnostic coordinate frame.

## Reporting summary

Further information on research design is available in the Nature Research Reporting Summary linked to this article.

## Data availability

All data analyzed in this paper were obtained from open-access sources. Disorder-specific Cohen's d maps derived from ENIGMA meta-analyses were accessed via the ENIGMA Toolbox (v. 1.1.3; https://enigma-toolbox.readthedocs.io/en/latest/;[43]). Through the toolbox, we also accessed normative connectivity data from a Human Connectome Project young adult sample (HCP; http://www.humanconnectome.org/;[45]), the von Economo-Koskinas cytoarchitectonic atlas[51], and gene transcriptomic data from the Allen human brain atlas (https://human.brain-map.org/) as accessible through Abagen (https://doi.org/10.5281/zenodo.4984124). The functional meta-analysis was based on the NeuroSynth database (https://neurosynth.org/). Developmental enrichment analyses were based on the Brainspan dataset (https://www.brainspan.org/static/download.html). Data generated for this study were made publicly available under https://github.com/CNG-LAB/cngopen/tree/main/transdiagnostic_gradients and https://doi.org/10.5281/zenodo.7180120. Raw imaging data supporting our findings are not publicly available as they contain information that could compromise the privacy of study participants. There are data sharing restrictions imposed by (i) ethical review boards of the participating sites, and consent documents; (ii) national and trans-national data sharing law, such as GDPR; and (iii) institutional processes, some of which require a signed MTA for limited and predefined data use. However, we welcome sharing data with researchers, requiring only that they submit an analysis plan for a secondary project to the leading team of the Working Group (http://enigma.ini.usc.edu). Once this analysis plan is approved, access to the relevant data will be provided contingent on data availability and local PI approval and compliance with all supervening regulations. If applicable, distribution of analysis protocols to sites will be facilitated. Source data are provided with this paper. Source data are provided with this paper (Supplementary Material). Source data are provided with this paper.

## Code availability

Custom code generated for this project was made publicly available under https://github.com/CNG-LAB/cngopen/tree/main/transdiagnostic_gradients and https://doi.org/10.5281/zenodo.7180120. Our analysis code makes use of open software: Gradient mapping analyses were carried out using BrainSpace (v. 0.1.2; https://brainspace.readthedocs.io/en/latest/) and epicenters were computed using code from the ENIGMA Toolbox (v. 1.1.3; https://enigma-toolbox.readthedocs.io/en/latest/;[43]). Visualizations were carried out using BrainStat (v. 0.3.6; https://github.com/MICA-MNI/BrainStat) in combination with ColorBrewer (v. 1.0.0; https://github.com/scottclowe/cbrewer2). Genetic analyses were performed using the GAMBA Toolbox (2021; https://github.com/dutchconnectomelab/GAMBA-MATLAB) and the cell-specific enrichment analysis tool (v. 1.1; http://genetics.wustl.edu/jdlab/csea-tool-2/).

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

## Acknowledgements

Many scientists around the world contributed to ENIGMA but did not take part in the writing of this report. A full list of contributors to ENIGMA is available at http:// enigma.ini.usc.edu/about-2/consortium/members/. The authors would like to express their gratitude to the open science initiatives that made this work possible: (i) the ENIGMA Consortium (core funding for ENIGMA was provided by the NIH Big Data to Knowledge (BD2K) program under consortium grant U54 EB020403 to P.M.T.), (ii) The Allen Human Brain Atlas and the abagen toolbox (https://doi.org/10.5281/zenodo.4984124), and (iii) the Human Connectome Project (principal investigators David Van Essen and Kamil Ugurbil; U54 MH091657), funded by the 16 NIH institutes and centers that support the NIH Blueprint for Neuroscience Research and by the McDonnell Center for Systems Neuroscience at Washington University. MDH was funded by the German Federal Ministry of Education and Research (BMBF) and the Max Planck Society. S.L. acknowledges funding from Fonds de la Recherche du Quebec – Sant (FRQ-S) and the Canadian Institutes of Health Research (CIHR). B.Y.P. was funded by the National Research Foundation of Korea (NRF-2021R1F1A1052303; NRF-2022R1A5A7033499), Institute for Information and Communications Technology Planning and Evaluation (IITP) funded by the Korea Government (MSIT) (No. 2022-0-00448, Deep Total Recall: Continual Learning for Human-Like Recall of Artificial Neural Networks; No. 2020-0-01389, Artificial Intelligence Convergence Research Center (Inha University); No. RS-2022-00155915, Artificial Intelligence Convergence Innovation Human Resources Development (Inha University); No. 2021-0-02068, Artificial Intelligence Innovation Hub), and Institute for Basic Science (IBS-R015-D1). M.H. is supported by a personal Veni grant from the Netherlands Organization for Scientific Research (NWO, grant number 91619115). J.B. is supported by the EU-AIMS (European Autism Interventions) and AIMS-2-TRIALS programmes which receive support from Innovative Medicines Initiative Joint Undertaking Grant No. 115300 and 777394, the resources of which are composed of financial contributions from the European Union's FP7 and Horizon2020 Programmes, and from the European Federation of Pharmaceutical Industries and Associations (EFPIA) companies' in-kind contributions, and AUTISM SPEAKS, Autistica and SFARI; and by the Horizon2020 supported programme CANDY Grant No. 847818). B.B. acknowledges research funding from the SickKids Foundation (NI17-039), the Natural Sciences and Engineering Research Council of Canada (NSERC; Discovery-1304413), CIHR (FDN-154298, PJT- 174995), the Azrieli Center for Autism Research (ACAR), an MNI-Cambridge collaboration grant, salary support from FRQ-S (Chercheur-Boursier), Brain-Canada, the Helmholtz International BigBrain Analytics and Learning Laboratory (Hiball) and the Canada Research Chairs (CRC) Program. S.L.V. was supported by the Max Planck Society through the Otto Hahn Award and the Helmholtz International BigBrain Analytics and Learning Laboratory (Hiball).

## Author contributions

The authors confirm contribution to the paper as follows: Study conception and design: M.D.H., S.L.V.; Data collection: O.A.C.D.H., L.S., O.A.A., C.R.K.C., M.H., K.B., D.V.R., D.J.V., D.J.S., B.F., T.G.M.V.E., N.J., P.M.T., S.I.T., ENIGMA ADHD Working Group, ENIGMA Autism Working Group, ENIGMA Bipolar Disorder Working Group, ENIGMA Major Depression Working Group, ENIGMA OCD Working Group, ENIGMA Schizophrenia Working Group; Analysis and interpretation of results: M.D.H., S.L.V.; Draft paper preparation: M.D.H., S.L.V.; Draft paper revision: M.D.H., S.B.E., B.C.B., P.M.T., S.I.T., R.A.I.B., S.L., B.Y.P., S.L.V.; All authors reviewed the results and approved the final version of the paper.

## Funding

## Competing interests

O.A.A. received speaker's honorarium from Lundbeck and Sunovion, Consultant to HealthLytix. Jan Buitelaar has been a consultant to/ member of advisory board of/and/or speaker for Takeda/Shire, Roche, Medice, Angelini, Janssen, and Servier. P.M.T. received grant support from Biogen, Inc., and consulting payments from Kairos Venture Capital, for work unrelated to the current paper. Other authors declare no competing interests.

## Additional information

M.D. Hettwer [1,2,3,4] ✉, S. Larivière[5], B. Y. Park [5,6,7], O. A. van den Heuvel [8], L. Schmaal [9,10], O. A. Andreassen [11], C. R. K. Ching[12], M. Hoogman[13], J. Buitelaar[14], D. van Rooij[14], D. J. Veltman[8], D. J. Stein [15], B. Franke [13], T. G. M. van Erp [16,17], ENIGMA ADHD Working Group*, ENIGMA Autism Working Group*, ENIGMA Bipolar Disorder Working Group*, ENIGMA Major Depression Working Group*, ENIGMA OCD Working Group*, ENIGMA Schizophrenia Working Group*, N. Jahanshad[12], P. M. Thompson[12], S. I. Thomopoulos[12], R. A. I. Bethlehem [18,19], B. C. Bernhardt[5], S. B. Eickhoff[1,3] & S. L. Valk [1,3,4] ✉

[1]Institute of Systems Neuroscience, Medical Faculty, Heinrich Heine University Düsseldorf, Düsseldorf, Germany. [2]Max Planck School of Cognition, Max Planck Institute for Human Cognitive and Brain Sciences, Leipzig, Germany. [3]Institute of Neuroscience and Medicine, Brain & Behavior (INM-7), Research Centre Jülich, Jülich, Germany. [4]Max Planck Institute for Human Cognitive and Brain Sciences, Leipzig, Germany. [5]Multimodal Imaging and Connectome Analysis Lab, McConnell Brain Imaging Centre, Montreal Neurological Institute, McGill University, Montreal, QC, Canada. [6]Department of Data Science, Inha University, Incheon, Republic of Korea. [7]Center for Neuroscience Imaging Research, Institute for Basic Science, Suwon, Republic of Korea. [8]Amsterdam UMC, Vrije Universiteit Amsterdam, Department of Anatomy and Neuroscience and Psychiatry, Amsterdam Neuroscience, Amsterdam, The Netherlands. [9]Centre for Youth Mental Health, The University of Melbourne, Melbourne, VIC, Australia. [10]Orygen, Parkville, VIC, Australia. [11]NORMENT Centre, Division of Mental Health and Addiction, University of Oslo and Oslo University Hospital, Oslo, Norway. [12]Imaging Genetics Center, Mark & Mary Stevens Neuroimaging and Informatics Institute, Keck School of Medicine, University of Southern California, Marina del Rey, CA, USA. [13]Departments of Psychiatry and Human Genetics, Donders Institute for Brain, Cognition and Behaviour, Radboud University Medical Center, Nijmegen, The Netherlands. [14]Department of Cognitive Neuroscience, Donders Institute for Brain, Cognition and Behaviour, Radboud University Medical Center, Nijmegen, The Netherlands. [15]South African Medical Research Council Unit on Risk & Resilience in Mental Disorders, Department of Psychiatry & Neuroscience Institute, University of Cape Town, Cape Town, South Africa. [16]Clinical Translational Neuroscience Laboratory, Department of Psychiatry and Human Behavior, University of California Irvine, Irvine Hall, Irvine, CA, USA. [17]Center for the Neurobiology of Learning and Memory, University of California Irvine, Irvine, CA, USA. [18]Autism Research Centre, Department of Psychiatry, University of Cambridge, Cambridge, UK. [19]Brain Mapping Unit, Department of Psychiatry, University of Cambridge, Cambridge, UK. *Lists of authors and their affiliations appear at the end of the paper ✉e-mail: m.hettwer@fz-juelich.de; s.valk@fz-juelich.de

## ENIGMA ADHD Working Group

M. Hoogman[13] & B. Franke [13]

## ENIGMA Autism Working Group

J. Buitelaar[14] & D. van Rooij[14]

## ENIGMA Bipolar Disorder Working Group

O. A. Andreassen [11] & C. R. K. Ching[12]

## ENIGMA Major Depression Working Group

D. J. Veltman[8] & L. Schmaal [9,10]

## ENIGMA OCD Working Group

O. A. van den Heuvel[8] & D. J. Stein [15]

## ENIGMA Schizophrenia Working Group

T. G. M. van Erp[16,17]

