## [Peer Review File · Nature Communications]

Coordinated cortical thickness alterations across six neurodevelopmental and psychiatric disordersREVIEWER COMMENTS

Reviewer #1 (Remarks to the Author):

Hettwer et al investigated the presence of coordinated cortical thickness alterations across 6 neuropsychiatric disorders using summary statistics from the ENIGMA consortium. Moreover, they contextualised their findings within patterns of organisation of the human brain, including macroscopical connectomics and transcriptomic data.

The manuscript is well written and the data well presented, the methods generally sound and the interpretation of the findings reasonable and well justified. I feel though that this manuscript would fit better within the remit of a more specialised journal.

This work did not generate any new datasets or tools and most of the findings are not particularly revealing in light of previous work conducted by the ENIGMA consortium itself and previous evidence demonstrating the apparent importance of connectomic and genetic features to regional vulnerability to pathology in the brain. Therefore, I regard this work as an incremental piece of evidence that brings these elements together with the study of shared morphological alterations across diagnosis boundaries. With the uprising interest in clinical heterogeneity (even within diagnosis boundaries), it is also not clear to me why this work might be particularly timely to fit in a wide-audience journal.

If this work is to objectively add something relevant to the literature, I would recommend the authors to invest some more time in thinking about what the concrete implications of these findings are and highlight this in the manuscript. Otherwise, I am afraid the message is not reaching the reader.

Reviewer #2 (Remarks to the Author):

Hettwer et al. mapped a transdiagnostic cortical thickness co-alteration network of 6 mental disorders by the aid of ENIGMA Meta-Analysis consortium. They further mapped the functional and structural epicenters of transdiagnostic cortical thickness alteration by refer to the normative functional and structural network from HCP. Then authors decomposed the co-alteration network to various transdiagnostic gradients of cortical thickness and associated the gradients to cytoarchitecture, gene expression and cognitive functions. The overall study is interesting and valuable for deepening the knowledge of neuro-structural foundations across mental disorders. The manuscript is well organized and written but with few issues need to be clarified. The detailed comments as follow.

1. The objective of this work is to identify the transdiagnostic cortical thickness co-alteration patterns of 6 mental disorders. But the selection of disorders is a bit inappropriate for this objective. Especially the co-alteration gradients shown disassociated patterns to different disorder's Cohen's d map (Table S3). What is the cross-disorders spatial correlation patterns of disease-specific Cohen's d maps (disorders x disorders)? Are there any sub-clusters to classify the disorders (like ADHD and OCD is cluster 1, BD, MDD and SCZ is cluster 2, ASD is cluster 3)? Can and how these gradients coordinate mental disorders?

2. The epicenter mapping approach to my understand is to characterize the potential region that its connectivity profile shows similar spatial representation with a disease effect. But in this manuscript, the co-alteration hubs do not necessarily represent high vulnerability to mental disorders which more like a high similarity of disease effect profiles. In other words, these hubs may be composed by several regions that show weak disease effects. So, what is the actual clinical significance of these co-alteration hubs and corresponding epicenters?

3. In fig S4, it looks like the transdiagnostic gradients without ADHD and ASD show more divergent pattern to the gradients in main results. Authors should quantify the similarity between these maps in fig S4 and the main one.

4. Does the underlying sample size or degree of freedom of different disease affects the co-alteration

network and the gradients?

5. Do authors have considered the effects spatial autocorrelation in “Genetic decoding” section? Within-category gene–gene coexpression and spatial autocorrelation are key drivers of the false-positive bias in gene-category enrichment analysis (GCEA) refer to Fulcher et al. The FDR correction approach alone cannot eliminate this false-positive bias and may lead to false results. Authors must take these effects into account in their analysis.

Fulcher, B. D., Arnatkeviciute, A. & Fornito, A. Overcoming false-positive gene-category enrichment in the analysis of spatially resolved transcriptomic brain atlas data. *Nat Commun* 12, 2669 (2021).

6. In page 9, “Associations with task-based functional activations” section, authors mentioned “We defined regions of interest as five-percentile bins of both gradients and studied the distribution of functional networks along the axes via z-statistics.” What is the meaning of this? How authors study the distribution of function networks is not clear and need more detailed description.

7. In Fig 2F, what the brain render in the left panel means? And the meaning of the color scheme in the scatter plot also not clear. What is the distribution of each cognitive term’s z-statistics looks like on gradient bins? Authors may provide more information about this section at least in SOM.

Minor:

1. The figures in this manuscripts and SOM have some parts lack for color bars and text marks, some of them (e.g., fig 2, S4, S6) may make reader confused.

2. The cortical surface renders should mask out the medial wall area given this work adopt a surface-based pipeline.

3. SOM, legend of Figure S4, typo “normalized angel”

Reviewer #3 (Remarks to the Author):

This paper examines the hypothesis that cortical alterations associated with psychiatric disorders covary in a biologically meaningful way across distinct diagnostic subgroups. This work follows on from recent studies that have investigated this hypothesis in individual diagnostic group – particularly schizophrenia. A strength of this paper is the inclusion of multiple diagnostic groups, allowing the authors to determine whether network-related cortical alterations are transdiagnostic. This is of particular importance as, while the disconnection hypothesis is well-supported for schizophrenia-spectrum disorders, there are fewer studies examining this hypothesis in other psychiatric disorders.

The paper is timely, well written and the various methodological approaches employed by the authors to test the hypothesis are sound. The formulated scientific question is of great interest to the neuroscientist community.

This is an excellent paper that deserves to be published. However, some aspects of the current version of the manuscript should be clarified and improved. Please find them listed hereafter.

It would be helpful to provide the sample sizes for each disorder in the main manuscript, so that the reader does not need to go to supplementary materials for this information.

In Figure 1a, it is difficult to ascertain which regions have positive d values due to the grey colour used. Would it be possible to use a colour that is easier to visualise? In addition, although correlation matrices are provided for the HCP data, it would be helpful to see a visualisation of network hubs projected onto brain images.

Could the authors please provide a justification for the inclusion of IQ as a covariate in their neuroimaging analyses for the ASD group. Similarly, was data for all bipolar, ASD, and ADHD participants collected at the same site? If not, is there a reason that site wasn’t included a a covariate for these samples?

Can the authors confirm that for the schizophrenia sample, all participants had a diagnosis of schizophrenia and not other schizophrenia-spectrum disorders? Previous studies have shown neuroimaging differences between schizophrenia-spectrum diagnoses (e.g. schizoaffective, schizophrenia, psychosis NOS, etc.), which may be important to consider in the current study.

What is missing for me is an examination of within-diagnosis covariance to supplement the transdiagnostic findings. Do the findings presented in Figure 1 hold across diagnostic groups, how much overlap is there between diagnoses? It is certainly interesting to see the transdiagnostic findings, however, I am left wondering what the similarities and differences might be across different diagnoses. Are cortical alterations driven by hubs of prominent covariance and epicentres in every disorder? Are findings the same for regions with reduced cortical thickness and those with increased cortical thickness?

Is it possible that the number of subjects across diagnoses could bias findings? I.e. could diagnostic groups with larger samples bias findings towards those groups? Is it possible to perform supplementary analyses with the number of subjects per group held constant at the same size to determine how sample size might impact results?

REVISION NCOMMS-22-06044

We would like to thank the Reviewers for their positive evaluations, constructive comments, and for the opportunity to submit a revised manuscript. We feel that the comments and suggestions have greatly improved our manuscript. In this cover letter, we outline the steps taken to address the suggestions of the Reviewers in a point-by-point fashion below and highlighted the corresponding changes in the manuscript in yellow.

REVIEWER COMMENTS

Reviewer #1 (Remarks to the Author):

Hettwer et al investigated the presence of coordinated cortical thickness alterations across 6 neuropsychiatric disorders using summary statistics from the ENIGMA consortium. Moreover, they contextualised their findings within patterns of organisation of the human brain, including macroscopical connectomics and transcriptomic data.

The manuscript is well written and the data well presented, the methods generally sound and the interpretation of the findings reasonable and well justified. I feel though that this manuscript would fit better within the remit of a more specialised journal.

This work did not generate any new datasets or tools and most of the findings are not particularly revealing in light of previous work conducted by the ENIGMA consortium itself and previous evidence demonstrating the apparent importance of connectomic and genetic features to regional vulnerability to pathology in the brain. Therefore, I regard this work as an incremental piece of evidence that brings these elements together with the study of shared morphological alterations across diagnosis boundaries. With the uprising interest in clinical heterogeneity (even within diagnosis boundaries), it is also not clear to me why this work might be particularly timely to fit in a wide-audience journal.

If this work is to objectively add something relevant to the literature, I would recommend the authors to invest some more time in thinking about what the concrete implications of these findings are and highlight this in the manuscript. Otherwise, I am afraid the message is not reaching the reader.

We thank the Reviewer for evaluating our work and are happy to further specify implications of these findings.

Our study suggests that mental disorders show coordinated brain alterations aligning with maturational, microstructural, functional and transcriptomic patterning across the cortex. We observed that in particular regions that show consistently shared disorder effects are part of similar co-alteration networks, and that these co-alteration networks are organized along axes aligning with micro- and macrostructural profiles. Such covariance axes are shaped by maturational sequences and shared genetic factors (Alexander-Bloch et al., 2019; Valk et al., 2020). Indeed, previous work has indicated differentiable spatial patterns of co-maturation and development along cortical axes, indicating the interplay of multiple neurodevelopmental mechanisms across the cortex (Fornito et al., 2019; Valk et al., 2020; Zhu et al., 2018). Our data-driven approach thus shines light on potential neurobiological mechanisms linked to maturational trajectories which underlie mental disorders.

Furthermore, our findings describe a coordinate system based on axes of illness effect covariance in which the impact of different mental disorders can be conceptualized. Such axes, describing smooth transitions of features across the cortex, have gained increasing interest in recent years. Organizational axes recapitulate macro-scale processing hierarchies across multiple neurobiological levels, including cortical macro- and microstructure, structure-function coupling, functional connectivity, cytoarchitecture, gene expression patterns, receptor architecture, and lifespan brain changes (see e.g. Bernhardt et al., 2022 for an overview). Seeing that normative brain development and cortical thickness covariance follow common organizational axes raises the question whether such axes also guide pathological changes causing mental illness. In addition, cortical alterations in mental disorders are assumed to be coordinated between regions throughout neurodevelopmental trajectories, rather than occurring in isolation (Dell’Osso et al., 2019; Menon, 2011). Thus, investigating shared brain alterations from a network perspective using methods that in part reflect shared maturation and genetic profiles is a promising new angle for transdiagnostic psychiatry.

Considering the uprising interest in clinical heterogeneity, we see our transdiagnostic approach as complementary to research on both disorder-specific features and within-disorder heterogeneity. That is, shared features crossing diagnostic boundaries are also a factor contributing to within-disorder heterogeneity and a patient’s position on a transdiagnostic spectrum. In the course of the revision, we also added a section on the embedding of individual disorders within this transdiagnostic space. Overall, we observed that disorders that show similar profiles of thickness impact are placed at similar positions along co-alteration axes, underscoring our concept of a transdiagnostic coordinate space. This implies that transdiagnostic covariance of disorder impact may describe a maturational organizational space in which individual disorder impact varies, i.e. that individual disorders are the result of the interplay of multiple neurodevelopmental processes uncovered by investigating their shared covariance.

We further elaborated on these aspects in the Discussion, p.18.

“Overall, our findings indicate that the degree to which regional alterations may be linked to and potentially facilitate alterations in other brain regions (i.e., potential epicenters), and the degree to which such interrelations pose a general feature of the mental disorders included (i.e., co-alteration hubs) appears to vary across the cortex and follows general neurobiological principles of brain organization (i.e., cortex-wide axes).

Last, we aimed to investigate how the proposed transdiagnostic co-alteration space, framed by both transdiagnostic covariance gradients, compares to previous descriptions of cross-disorder similarities and disorder clusters. That is, our cross-disorder covariance approach generates a continuous space within which disorders vary with respect to their topography of similarity to transdiagnostic patterns across the cortex. While we indeed found that positions of disorders within this space converge with their allocation to disorder clusters, the co-alteration framework captures both similarities within and between clusters in a continuous manner. By embedding illness effects within a space shaped by genetic and maturational processes, we gain further insight in differentiable neurobiological mechanisms underlying individual disorders. Indeed, the first gradient, stretching between frontal and temporal regions showed similarities with a previously described anterior-posterior axis along the cortical mantle (Valk, 2020). Previous work has indicated differentiable spatial patterns of co-maturation and development along various spatial axes, indicating the interplay of multiple neurodevelopmental mechanisms across the cortex (Fornito et al., 2019; Valk et al., 2020; Zhu et al., 2018). The observed systematic alterations along such axes across disorders may reflect

differential disruptions in pre- and post-natal neurodevelopment. Moreover, we observed that, for most disorders, overlap between disorder-specific and transdiagnostic covariance is highest in heteromodal cortices. This convergence may be linked to their placement within these neurodevelopmental axes (Baum et al., 2021; Paquola et al., 2019), supporting these regions as targets of transdiagnostic investigations. Future work may evaluate potential causes and critical time windows of development within this framework, enhancing our understanding of the ontogeny of cortical organization in health and disorder.”

[...] p. 20:

“While our findings underline the relevance of transdiagnostic approaches, they do not contradict the existence of etiological and phenomenological differences within and between psychiatric diagnoses. Our transdiagnostic approach does not capture heterogeneity within and between highly related diagnostic categories, as expected to be present e.g. within the included SCZ (schizophrenia spectrum disorders) and BD (type I and II combined) sample. However, shared features crossing diagnostic boundaries are likely also an important factor contributing to within-disorder heterogeneity. Moreover, individuals may be diagnosed with multiple different disorders across their lifespan (Plana-Ripoll et al., 2019). Understanding which neurobiological principles drive the spectrum of varying mental disorders is a crucial piece of the puzzle of the biological origin of disorder variability. Thus, investigating both disorder-specific phenomena and heterogeneity within (spectrum-)diagnoses forms a crucial line of research that will continue to complement our transdiagnostic findings. Presented cortex-wide co-alteration features shall facilitate and provide a new transdiagnostic coordinate frame for such insights.”

Reviewer #2 (Remarks to the Author):

Hettwer et al. mapped a transdiagnostic cortical thickness co-alteration network of 6 mental disorders by the aid of ENIGMA Meta-Analysis consortium. They further mapped the functional and structural epicenters of transdiagnostic cortical thickness alteration by refer to the normative functional and structural network from HCP. Then authors decomposed the co-alteration network to various transdiagnostic gradients of cortical thickness and associated the gradients to cytoarchitecture, gene expression and cognitive functions. The overall study is interesting and valuable for deepening the knowledge of neuro-structural foundations across mental disorders. The manuscript is well organized and written but with few issues need to be clarified. The detailed comments as follow.

R2.Q1a. *The objective of this work is to identify the transdiagnostic cortical thickness co-alteration patterns of 6 mental disorders. But the selection of disorders is a bit inappropriate for this objective. Especially the co-alteration gradients shown disassociated patterns to different disorder's Cohen's d map (Table S3). What is the cross-disorders spatial correlation patterns of disease-specific Cohen's d maps (disorders x disorders)? Are there any sub-clusters to classify the disorders (like ADHD and OCD is cluster 1, BD, MDD and SCZ is cluster 2, ASD is cluster 3)?*

We thank the Reviewer for these comments. There were several reasons that led us to include specifically these six disorders in our transdiagnostic study. Schizophrenia (SCZ), bipolar disorder (BD), major depressive disorder (MDD), obsessive-compulsive disorder (OCD), attention-deficit/hyperactivity disorder (ADHD), and autism spectrum disorder (ASD) are widely studied psychiatric conditions for which collaborative efforts of the ENIGMA consortium have advanced research on disease-related brain alterations in sufficiently large samples with increased statistical power. These six disorders have been found to show overlap on multiple levels, including clinical symptoms, phenotypic brain alterations, and (genetic) risk factors (e.g. (Insel et al., 2010; Lee et al., 2019; Marshall, 2020; Opel et al., 2020; Park et al., 2021; Patel et al., 2021; Radonjic et al., 2021). Moreover, the selection of these six disorders conforms to previous transdiagnostic studies (Opel et al., 2020; Park et al., 2021; Patel et al., 2021; Radonjic et al., 2021). Radonjic et al. additionally included epilepsy, but reported low or no phenotypic and genetic similarity to all of the six mentioned mental disorders. Thus, we did not include epilepsy. We also excluded the 22q11.2 deletion syndrome though ENIGMA summary statistics would have been available, as it is not a psychiatric diagnosis per se that is included in the DSM-V but rather a condition that increases risk for (other) psychiatric conditions (and also includes other anomalies such as cardiac defects, craniofacial anomalies or intellectual disability). By staying consistent with previous transdiagnostic ENIGMA studies and including the same selection of disorder we can directly extend previous insights on shared spatial patterns of illness effects. At the same time, we acknowledge that availability of ENIGMA cortical thickness maps for an even wider range of psychiatric conditions would draw a broader picture, which we now further emphasize in the Discussion of the Manuscript.

We elaborated on the disorder selection in the beginning of the Results section, p. 5.

“Consistent with previous work (e.g. Patel et al., 2021; Opel et al., 2020; Park et al., 2021) we selected six mental disorders for which illness effects have been studied in large samples in collaborative international meta-analyses by the ENIGMA consortium.”

and Discussion, p. 21.

“It should further be noted that also other disorders such as substance abuse or anxiety disorders tend to co-occur with some of the disorders included here, but could not be included in the analyses as ENIGMA cortical thickness summary statistics have not yet been published. Further work including a wider range of disorders will help to evaluate the generalizability of our transdiagnostic model.”

In the current work, we used a covariance approach to study the shared impact on cortical thickness by the six disorders. Though indeed co-alteration gradients show differential patterns to different disorders, this does not undermine our results. Rather, it highlights that disorders vary in their impact within this ‘transdiagnostic space’. Previous ENIGMA papers have reported cross-disorder correlation matrices (see e.g. Radonjic et al., 2021; Opel et al., 2020). Opel et al. describe a certain degree of clustering, where schizophrenia (SCZ), bipolar disorder (BD), major depressive disorder (MDD), and obsessive-compulsive disorder (OCD) show different patterns of disease effects compared to attention-deficit/hyperactivity disorder (ADHD) and autism spectrum disorder (ASD). We now additionally confirmed and depicted this in a dendrogram together with a cross-disorder correlation matrix (see **Fig. 3C&D**) to aid direct comparison of approaches. In contrast to clustering approaches, our cross-disorder covariance approach aims to describe a continuous transdiagnostic space in which disorder effects occur. Notably, the degree to which cortical alterations are shared between disorders varies across the cortex. Similarly, the degree to which cortical alteration patterns in individual disorders align with transdiagnostic patterns varies across the cortex, along axes of cross-disorder covariance. We therefore take an integrative view on the spatial and transdiagnostic embedding of regional illness effects, where various features (co-alteration hubs, disease epicenters, organizational axes) capture convergent information on systematic interrelationships of regional pathological alterations.

We find that disorders which cluster together, such as SZC, BD, and OCD, show a similar placement within this transdiagnostic co-alteration framework (see **Fig. 3D-F**). At the same time, we also observe disorders that show some similarity but are allocated to different clusters, such as MDD and ADHD, to be positioned closer to each other in our transdiagnostic space. This continuous approach thus appears to capture shared patterns across cluster boundaries in line with the notion of overlapping clinical spectra. Here, the similarity between individual disorders is captured independently from the similarity between other disorders which would additionally influence cluster allocation. This aspect appears especially relevant in the example of MDD and ADHD, which occur frequently as comorbidities (Katzman et al., 2017) and show significant phenotypic similarity (correlation between ENIGMA maps: $r = 0.44$). However, clustering allocates them to different clusters as they show different phenotypic overlaps with the other four disorders. Aiming to investigate factors that characterize shared features of mental disorders, our concept of a transdiagnostic coordinate space thus highlights how disorders with similar profiles of thickness impact are placed at similar positions along co-alteration axes.

We added this aspect to the Results and Discussion sections in combination with points raised in #R2.Q1b. Thus, please see the respective changes below.

R2.Q1b. Can and how these gradients coordinate mental disorders?

The first transdiagnostic gradient, stretching between frontal and temporal regions, partly mirrors previous findings on normative structural covariance gradients that are shaped by maturational sequences and shared genetic factors (Alexander-Bloch et al., 2019; Valk et al., 2020). Individual disorders vary along transdiagnostic axes, potentially according to their pathological trajectories during

development and prominent vulnerable developmental windows. Indeed, we observed differences in co-alteration network embedding in regions with different cytoarchitectonic and functional profiles. Knowing that functional networks, e.g., sensory compared to higher level association, mature during different developmental stages, these likely also overlap to different degrees with vulnerable periods of psychiatric disorders. Thus, by embedding illness effects within a space shaped by genetic and maturational processes, we gain further insight in differentiable neurobiological mechanisms underlying disorders. These are anchored in neurobiological gradients, beyond insights given by clustering alone. To make this point clearer, we now display the individual embedding of disorder effects along transdiagnostic gradients in our main Results, p. 12.

“Embedding of individual disorders within a transdiagnostic co-alteration space

“Next, we compared the degree of similarity between disorders and their embedding within the transdiagnostic co-alteration space. Replicating what previous transdiagnostic studies have shown (Radonjic et al., 2019; Opel et al., 2020), we observed highest similarity between SCZ, BD, and OCD, while ADHD and ASD form a separate cluster (Fig. 3C&D). In contrast to clustering approaches, our cross-disorder covariance approach aimed to describe a transdiagnostic organizational space in which disorder effects occur. Indeed, we found that disorders that cluster together, such as SCZ, BD, and OCD show a similar placement within this transdiagnostic co-alteration framework (see Fig. 3D-F). While transdiagnostic hubs correlated with illness effect maps in SCZ ($r = 0.76$, $p_{spin} < .0001$), BD ($r = 0.66$, $p_{spin} = 0.001$), and OCD ($r = 0.26$, $p_{spin} = 0.03$), this was not the case for ASD ($r = 0.07$, $p_{spin} = 0.31$) and MDD ($r = -0.05$, $p_{spin} = 0.41$), and ADHD showed a negative correlation ($r = -0.42$, $p_{spin} = 0.003$). Similarly, disorder-specific epicenters overlapped with transdiagnostic epicenters in SCZ, BD, and OCD, and in part in MDD (see Fig. 3E), whereas ADHD and ASD showed no significant disorder-specific epicenters in the first place. That is, illness effect maps relate to similar degrees to transdiagnostic co-alteration hubs, are linked to epicenters that overlap to similar degrees with transdiagnostic epicenters (Fig. 3E), and are positioned more closely together in a transdiagnostic covariance space framed by G1 and G2 (Fig. 3F). However, we also observe disorders that show some similarity but are allocated to different clusters, such as MDD and ADHD, to be positioned closer to each other in our continuous transdiagnostic space, crossing cluster boundaries. Overall, disorders that show similar profiles of thickness impact are placed at similar positions within a transdiagnostic coordinate space.”

and Discussion, p. 19.

“By embedding illness effects within a space shaped by genetic and maturational processes, we gain further insight in differentiable neurobiological mechanisms underlying individual disorders. Indeed, the first gradient, stretching between frontal and temporal regions showed similarities with a previously described anterior-posterior axis along the cortical mantle (Valk, 2020). Previous work has indicated differentiable spatial patterns of co-maturation and development along multiple spatial axes, indicating the interplay of multiple neurodevelopmental mechanisms across the cortex (Fornito et al., 2019; Valk et al., 2020; Zhu et al., 2018). The observed systematic alterations along such axes across disorders may reflect differential disruptions in pre- and post-natal neurodevelopment. Future work may evaluate potential causes and critical time windows of development within this framework, enhancing our understanding of the ontogeny of cortical organization in health and disorder.”

Figure 3. Embedding of six disorders within transdiagnostic co-alteration networks. [...] **C)** Similarity of illness effects between disorders, i.e., correlations of Cohen's d maps, and how they cluster together (**D**). Position of individual disorders within a transdiagnostic co-alteration space based on **E**) the correlation between transdiagnostic hubs and Cohen's d maps (x-axis) and the overlap between transdiagnostic and disorder-specific epicenters (y-axis); and **F**) the correlation between transdiagnostic gradients G1 and G2 and Cohen's d maps.

R2.Q2. *The epicenter mapping approach to my understand is to characterize the potential region that its connectivity profile shows similar spatial representation with a disease effect. But in this manuscript, the co-alteration hubs do not necessarily represent high vulnerability to mental disorders which more like a high similarity of disease effect profiles. In other words, these hubs may be composed by several regions that show weak disease effects. So, what is the actual clinical significance of these co-alteration hubs and corresponding epicenters?*

We thank the Reviewer for this comment. As previous work has indicated, epicenter mapping aids to understand how the normative connectivity profile associated with a specific region may play a central role in the manifestation of a disorder (Larivière et al., 2020; Shafiei et al., 2020; Zhou et al., 2012). Contrary to network hubs, epicenters do not necessarily need to be regions most strongly affected, but rather play a role in shaping co-alteration patterns due to their network-embedding. In our case, epicenters reflect regions whose connectivity profile may underlie illness effects that are especially strongly organized across disorders (i.e., in co-alteration hubs). Thus, disease epicenter mapping underscores how patterns of transdiagnostic effects are anchored in the connectivity of structural and functional sub-networks.

We have further explained this in the Results, p. 6.

“As previous work has indicated, epicenter mapping aids to understand how the normative connectivity profile associated with a specific region may play a central role in the manifestation of a disorder (Zhou, 2012; Shafiei, 2020; Lariviere, 2020). Here, we identified transdiagnostic epicenters as regions whose connectivity profile may underlie illness effects that are consistently organized across disorders, i.e. regions whose network embedding correlates significantly with co-alteration hubs. Thus, the epicenter mapping approach highlights the role of regions that do not necessarily constitute hubs themselves (Lariviere, 2020) but may contribute to shaping shared patterns of illness effects through strong or distributed connections with co-alteration hubs.”

The aim of the current study is to investigate how illness effects are systematically organized across the cortex, which spans both “coordinated impact” as well as “coordinated retainment”.

However, we agree that interpretation of our results would be more intuitive with additional knowledge on what co-alteration hubs reflect. To address this concern, we performed an additional analysis focussing on regions that are most strongly and consistently altered across disorders (primarily heteromodal regions), or consistently unaffected across disorders (primary unimodal and paralimbic regions). When studying the sum of illness effect maps (using absolute Cohen's d values and re-scaling within disorder maps to account for offsets in overall illness effects) highlighting regions consistently affected across disorders, we observed a significant correlation with transdiagnostic co-alteration hubs ($r = 0.42$, $p_{spin} < .0001$). This indicates that hubs are indeed placed primarily in regions with shared impact. Correspondingly, when computing the epicenters based on the mean hit map, we again identified transdiagnostic epicenters primarily in prefrontal and temporal regions. These epicenters overlapped highly with transdiagnostic epicenters computed based on the co-alteration hubs (see **Supplementary Fig. S1E&F** below). We have now included this additional information in the Results, p. 5.

“When studying which regions are most strongly and consistently affected across disorders via the sum of normalized illness effect maps (see Supplementary Fig. S1B), we observed a significant correlation with transdiagnostic co-alteration hubs ($r = 0.42$, $p_{spin} < .0001$), suggesting hubs are placed in regions with shared impact.”

[...] p. 6

“Systematically investigating connectivity profiles of 68 cortical seeds revealed primarily temporal and prefrontal regions as potential transdiagnostic disease epicenters (Fig. 1D). This finding held true when computing epicenters based on the ‘hit map’ (Supplementary Fig. S1E&F).“

And as **Supplementary Fig. S1**.

Figure S1. Correspondence between co-alteration hubs, shared illness effects, and epicenters. **A)** Co-alteration hubs; **B)** Hit map based on average absolute Cohen's d values, rescaled between 0 and 1 within disorders. Overlaps in cortical thickness (CT) reductions (**C**) and increase (**D**). In subplots **A - D**), black and white brain images show a thresholded version (top 20%) of the brain image in the same subplot. **E)** and **F)** depict disease epicenters computed based on the hit map (**B**) for functional and structural connectivity, respectively.

We observed that in particular regions that are most strongly affected in most disorders show similar co-alteration patterns across the cortex. Investigating how pathological alterations in cortical thickness are interrelated among some but not other regions yields new insights into the topography of shared and distinguishable processes among mental disorders. That is, illness effects in some regions, particularly temporal and prefrontal heteromodal cortices, are not only shared consistently across disorders but are also synchronized similarly with illness effects in other cortical regions.

In conclusion, the degree to which regional alterations may be linked to and potentially facilitate alterations in other brain regions, (i.e., potential epicenters) and the degree to which such interrelations pose a general feature of the mental disorders included (i.e., co-alteration hubs) varies across the cortex and follows general neurobiological principles of brain organization (i.e., cortex-wide gradients). As the current work was based on summary scores, we unfortunately did not have access to clinical features associated with each disorder, making inferences about clinical relevance limited. At the same time, understanding the principles according to which cortical alteration patterns are organized across diagnoses will provide a fruitful basis for further investigations on the interrelationship between

network organization and symptoms shared across disorders, as well as variations within categorical diagnoses.

We have now outlined these considerations further in the Discussion, p. 15

“We identified hubs of transdiagnostic co-alteration predominantly in lateral and ventral temporal lobes, with some impact on post-central and medial frontal regions. Importantly, these hubs overlapped with regions showing shared thickness alterations, indicating especially pronounced coordination within co-alteration networks between consistently affected regions. Observing an interrelationship of pathological cortical thickness alterations between temporal and prefrontal heteromodal cortices, but less so in unimodal and paralimbic cortices, indicates distinguishable processes shared between disorders and across the cortex.”

[...]

p. 15

“That is, transdiagnostic epicenters suggest a central role of prefrontal and temporal cortex in the manifestation of mental illness, indicating how transdiagnostic cortical alterations are anchored in the connectivity of identified regions.”

[...]

p. 18

“Overall, our findings indicate that the degree to which regional alterations may be linked to and potentially facilitate alterations in other brain regions (i.e., potential epicenters), and the degree to which such interrelations pose a general feature of the mental disorders included (i.e., co-alteration hubs) appears to vary across the cortex and follows general neurobiological principles of brain organization (i.e., cortex-wide axes).”

[...]

p. 21

“Last, the lack of subject-level clinical information impeded the direct assessment of clinical implications of current findings. However, understanding the principles according to which cortical alteration patterns are organized across diagnoses will provide a fruitful basis for further investigations on the interrelationship between network organization and symptoms shared across disorders, as well as variations within categorical diagnoses.”

R2.Q3. In fig S4, it looks like the transdiagnostic gradients without ADHD and ASD show more divergent pattern to the gradients in main results. Authors should quantify the similarity between these maps in fig S4 and the main one.

We thank the Reviewer for noting this and are happy to further clarify. In the supplementary section “Influence of individual disorders on gradient organization“, we provide quantification of the similarity between maps shown in Fig. S4 (of the initial submission) and the original transdiagnostic gradients G1 and G2. In our revised manuscript, we further clarified this figure and included it in a general figure on robustness to parameter manipulations for a faster and clearer overview (see supplementary **Fig. S2B - right panel**).

Figure S2. Robustness of co-alteration hubs and transdiagnostic gradients to parameter manipulations. Values indicate correlation with original hubs/gradients after parameter manipulation. **A)** Co-alteration hubs based on co-alteration matrix with different cut-offs or corrected for sample-size (n-corrected) per disorder. **B)** Left: Corrected for average illness effects and sample size, or Laplacian eigenmap or principal component analysis (PCA) as dimensionality reduction techniques, or co-alterations based on spearman's rho. Middle: Co-alteration matrix cut-offs. Right: Constructing gradients based on five disorders only, highlighting the contribution of single disorders. *G1 and G2 are switched for autism spectrum disorder (ASD). BD = Bipolar disorder, ADHD = Attention-deficit/hyperactivity disorder, MDD = Major depressive disorder, OCD = Obsessive compulsive disorder, SCZ = Schizophrenia spectrum disorder.

Indeed, leaving out ASD in the gradient computation leads to a divergence in observed patterns. It appeared to lead to a switch in features to be reflected in principal and secondary gradients, as was observed in significant correlations of the principal gradient without ASD with the original G2 ($r = 0.86$, $p_{spin} < .001$) and of the secondary gradient without ASD with the original G2 ($r = 0.48$, $p_{spin} < 0.01$). This finding is not surprising, as cortical thickness alterations in ASD show a spatial pattern that is highly similar to G1 and thus likely strengthens the weight of features then represented in G1." We have now further highlighted this in the supplementary section "Influence of individual disorders on gradient organization".

"leaving out ASD in gradient computations appeared to lead to a switch in features to be reflected in principal and secondary gradients, as was observed in significant correlations of the principal gradient without ASD with the original G2 ($r = 0.86$, $p_{spin} < .001$) and of the secondary gradient without ASD with the original G2 ($r = 0.48$, $p_{spin} < 0.01$). This finding is not surprising, as cortical thickness alterations in ASD show a spatial pattern that is highly similar to G1 and thus likely strengthens the weight of features then represented in G1."

4. Does the underlying sample size or degree of freedom of different disease affects the co-alteration network and the gradients?

Thank you for noting this. In theory, the differences in sample size are already accounted for to a certain degree, as summary statistics are given as Cohen's d values, which is a standardized effect size measure designed to allow comparison between studies and including the sample size as a variable in its formula.

In addition, we now evaluated the impact of including sample size as a covariate in the computation of the cross-disorder correlation matrix that hub and gradient analyses are based on.

Correcting for sample sizes did not substantially influence the results:

- Correlation of n-corrected co-alteration hub map with original co-alteration hub map: $r = 0.89$, $p_{spin} < .0001$
- Gradient 1: $r = 0.99$, $p_{spin} < .0001$
- Gradient 2: $r = 0.96$, $p_{spin} < .0001$

We included these additional quality checks in the Results sections *Transdiagnostic covariance hubs inform disease epicenters*, p. 6

“Findings were comparable at different thresholds and when correcting for sample size (see Fig. S2).”

and *Macroscale gradients of transdiagnostic co-alteration networks*, p. 8

“Findings were comparable at different thresholds and robust against parameter manipulation, sample size correction, and selection of diagnoses (see Fig. S2).”

as well as more detailed in the supplementary sections *“Robustness of cross-disorder co-alteration network hubs”* and *“Robustness of transdiagnostic gradients”*.

“Correcting the cross-disorder co-alteration matrix for sample sizes via a partial correlation did not impact resulting hubs (correlation of n-corrected co-alteration hub map with original co-alteration hub map: $r = 0.89$, $p_{spin} < .0001$; Fig. S2A).”

[...]

“Fourth, gradients were not impacted by correcting for sample sizes of underlying disorder samples as tested by including sample sizes as covariates in the computation of the cross-disorder co-alteration matrix via partial correlation (G1: $r = 0.99$, p_{spin} ; G2: $r = 0.96$, p_{spin} ; Fig. S2B).”

Figure S2. Robustness of co-alteration hubs and transdiagnostic gradients to parameter manipulations. Values indicate correlation with original hubs/gradients after parameter manipulation. **A)** Co-alteration hubs based on co-alteration matrix with different cut-offs or corrected for sample-size (n-corrected) per disorder. **B)** Left: Corrected for average illness effects and sample size, or Laplacian eigenmap or principal component analysis (PCA) as dimensionality reduction techniques, or co-alterations based on spearman's rho. Middle: Co-alteration matrix cut-offs. Right: Constructing gradients based on five disorders only, highlighting the contribution of single disorders. *G1 and G2 are switched for autism spectrum disorder (ASD). BD = Bipolar disorder, ADHD = Attention-deficit/hyperactivity disorder, MDD = Major depressive disorder, OCD = Obsessive compulsive disorder, SCZ = Schizophrenia spectrum disorder.

R2.Q5. Do authors have considered the effects spatial autocorrelation in “Genetic decoding” section? Within-category gene–gene coexpression and spatial autocorrelation are key drivers of the false-positive bias in gene-category enrichment analysis (GCEA) refer to Fulcher et al. The FDR correction approach alone cannot eliminate this false-positive bias and may lead to false results. Authors must take these effects into account in their analysis.

Fulcher, B. D., Arnatkeviciute, A. & Fornito, A. Overcoming false-positive gene-category enrichment in the analysis of spatially resolved transcriptomic brain atlas data. *Nat Commun* 12, 2669 (2021).

We thank the reviewer for pointing out this indeed very relevant aspect and included testing for spatial and gene specificity in the revised manuscript. First, we followed recommendations from Fulcher et al. (2021) and Arnatkeviciute et al. (2022) suggesting the use of ensemble-based null models for neuroimaging-transcriptomics association analyses. To this end, we generated a set of random spatially autocorrelated phenotype maps (Alexander-Bloch et al., 2018) to test the spatial specificity of associations observed between gene transcriptomic profiles and transdiagnostic gradients. Genes with an expression profile significantly correlated with the primary transdiagnostic gradient (G1; $p_{spin} < .01$) were defined as gene set (n = 232 genes) for following gene specificity tests. Using the Gene Annotation using Macroscale Brain-imaging Association (GAMBA) Toolbox (Wei et al., 2022), we tested this gene set against two types of null models: The null-coexpressed-gene model and the null-brain-gene model. The null-coexpressed-gene model includes genes with a similar co-expression level as the gene set of interest to generate null distributions. The null-brain-gene model generates null models exclusively from genes over-expressed in brain tissue and is thus more conservative than classical random-gene models. The set of 232 genes showing a spatially specific association to G1 also showed gene specificity

(null-coexpressed-gene model: $p < .0001$; null-brain-gene model: $p < .0001$). We updated this information in the manuscript accordingly.

Results p. 9.

“We generated null models to assess spatial specificity (including spatially autocorrelated phenotype maps) and gene specificity (including a) genes with similar levels of coexpression and b) genes overexpressed in the brain compared to the rest of the body) of the identified gene set.”

and Methods, p. 26.

“We correlated transdiagnostic gradients with the post-mortem gene expression maps and tested for spatial and gene specificity using several null models: First, we generated a set of random spatially autocorrelated phenotype maps (Alexander-Bloch, 2018) to test the spatial specificity of associations observed between gene transcriptomic profiles and transdiagnostic gradients. Genes with an expression profile significantly correlated with G1 or G2 ($p_{spin} < .01$) were defined as gene set for following gene specificity tests. Next, using the Gene Annotation using Macroscale Brain-imaging Association (GAMBA) Toolbox (Wei, 2022), we tested this gene set against two types of null models: The null-coexpressed-gene model and the null-brain-gene model. The null-coexpressed-gene model includes genes with a similar co-expression level as the gene set of interest to generate null distributions. The null-brain-gene model generates null models exclusively from genes over-expressed in brain tissue and is thus more conservative than classical random-gene models. If a gene set was identified as significantly associated with a transdiagnostic gradient in both linear regressions and described permutation tests, it was next used as input for a developmental enrichment analysis via the cell-type specific expression analysis (CSEA) developmental expression tool (<http://genetics.wustl.edu/jdlab/csea-tool-2>).”

and **Figure 2E**.

Figure 2. Macroscale organization of transdiagnostic covariance in cortical thickness alterations. [...] **E)** Developmental gene enrichment analysis based on 232 genes for which spatial expression patterns correlated with G1 (of which 146 showed a positive correlation, i.e., were overexpressed in prefrontal compared to temporal regions). [...]

R2.Q6. In page 9, “Associations with task-based functional activations” section, authors mentioned “We defined regions of interest as five-percentile bins of both gradients and studied the distribution of functional networks along the axes via z-statistics.” What is the meaning of this? How authors study the distribution of function networks is not clear and need more detailed description.

We are happy to further clarify. To evaluate the association between gradient loadings and functional networks we used the NeuroSynth meta-analytical framework, as previously adopted in Margulies et al. (2016) and Paquola et al. (2019). We re-phrased the explanation, using the term “position along gradients” rather than “distribution”, and expanded the respective Results section, p.10.

“We binned each gradient into five-percentile bins and defined regions of the same bin as a region of interest (ROI). Resulting 20 ROIs for each gradient were then tested for their overlap with meta-analytic ROIs associated with each of the 24 cognitive terms via z-statistics. The magnitude of an average z-value at a ROI (i.e., a position along the gradient) reflects the strength of its association with a certain functional task activation. We sorted the topic terms by their weighted mean position along both gradients, revealing systematic shifts in functional networks along transdiagnostic axes of co-alteration.”

and Methods section, p.27.

“This analysis adhered to procedures previously demonstrated by Margulies et al. (2016) and Paquola et al. (2019). Briefly, we derived 20 ROI maps by decomposing G1 and G2 into five-percentile bins and combining regions of the same bin to a joint ROI. The granularity of five-percentile bins is assumed to capture subtle variations along cortical axes. We then examined the association of each ROI with 24 cognitive topic terms via z-statistics.”

R2.Q7. In Fig 2F, what the brain render in the left panel means? And the meaning of the color scheme in the scatter plot also not clear. What is the distribution of each cognitive term’s z-statistics looks like on gradient bins? Authors may provide more information about this section at least in SOM.

The brain render of the left panel reflected a combination of G1 and G2 loadings, with colors within their shared axes noted in the scatter plot. However, on second thought we agree that the brain plot and color scheme may be confusing to the reader and are not central to understanding the message. Therefore we excluded both in the revised manuscript. Moreover, we included the distribution of each cognitive term’s z-statistic for each gradient separately in the supplementary material. Combined with the information added in the main text as a response to comment #R2.Q6, we hope that our analysis and results are now presented more clearly.

We added the following section to the Supplementaries:

“Distribution of NeuroSynth functional terms along gradient bins

In addition to the 2D space framed by the two transdiagnostic gradients within which cognitive terms were situated, see Fig. 2 of the main manuscript, we also investigated the position of cognitive terms along each gradient separately. Following the same strategy and using the same 24 NeuroSynth cognitive terms, we binned each gradient into five-percentile bins. Regions of the same bin formed a region of interest (ROI), yielding 20 ROIs for each gradient. These ROIs were then tested for their overlap with meta-analytic ROIs associated with each of the 24 cognitive terms via z-statistics. The magnitude of an average z-value at a ROI (i.e., a position along the gradient) reflects the strength of its association with a certain functional task

activation. We sorted the topic terms by their weighted mean position along both gradients, revealing systematic shifts in functional networks along transdiagnostic axes of co-alteration (Fig. S6). While G1 segregates sensory ('auditory', 'multisensory') from higher order cognitive functions ('Cognitive control', 'inhibition'), G2 distinguishes sensory ('auditory', 'multisensory') from perception/attention related functions ('visuospatial', 'attention')."

Figure S6. Meta-analysis for 24 cognitive terms obtained from NeuroSynth along the principal (G1) and secondary gradient (G2). We computed parcel-wise z-statistics, capturing node-function associations, and calculated the center of gravity of each function along 20 five-percentile bins of G1 (A) and G2 (B). Function terms are ordered by the weighted mean of their location along the gradients.

Minor:

R2.Q8. The figures in this manuscripts and SOM have some parts lack for color bars and text marks, some of them (e.g., fig 2, S4, S6) may make reader confused.

We thank the Reviewer for this comment and apologize for the confusion. We have now adjusted the figures and corresponding color bars accordingly, please find updated figures below and in the manuscript. Note that we removed Fig. S4 and integrated it with a general figure on robustness to parameter manipulations for a faster and clearer overview (see supplementary Fig. S2B). Fig. S6 is now Fig. S4.

Figure 2. Macroscale organization of transdiagnostic covariance in cortical thickness alterations. **A)** A cross-condition structural covariance matrix was thresholded at 80% and decomposed using diffusion map embedding. Covariance along the principal (G1) and second (G2) gradients is depicted on the right. **B)** Transdiagnostic gradients G1 and G2. **C)** Correlation between a normative axis of cortical thickness covariance and transdiagnostic gradients. **D)** Cross-condition gradients stratified according to von Economo-Koskinas cytoarchitectonic classes. **E)** Developmental gene enrichment analysis based on 232 genes for which spatial expression patterns correlated with G1 (of which 146 showed a positive correlation, i.e., were overexpressed in prefrontal compared to temporal regions). **F)** Meta-analysis for diverse cognitive functions obtained from NeuroSynth similar to Margulies et al. . We computed parcel-wise z-statistics, capturing node-function associations, and calculated the center of gravity of each function along 20 five-percentile bins of G1 and G2. Function terms are ordered by the weighted mean of their location along the gradients.

Figure S4. *Gradient loadings at epicenters.* Principal axis (G1) masked by significant functional epicenters, demonstrating that epicenters are strongly placed towards apices of the gradient. Red and blue colors indicate opposite apices of G1.

R2.Q9. *The cortical surface renders should mask out the medial wall area given this work adopt a surface-based pipeline.*

Thanks for the suggestion. We now mask out the medial wall for all brain plots.

R2.Q10. *SOM, legend of Figure S4, typo “normalized angel”*

We removed this figure and included it in a general figure on robustness to parameter manipulations (Fig. S2).

Reviewer #3 (Remarks to the Author):

This paper examines the hypothesis that cortical alterations associated with psychiatric disorders covary in a biologically meaningful way across distinct diagnostic subgroups. This work follows on from recent studies that have investigated this hypothesis in individual diagnostic group – particularly schizophrenia. A strength of this paper is the inclusion of multiple diagnostic groups, allowing the authors to determine whether network-related cortical alterations are transdiagnostic. This is of particular importance as, while the disconnection hypothesis is well-supported for schizophrenia-spectrum disorders, there are fewer studies examining this hypothesis in other psychiatric disorders.

The paper is timely, well written and the various methodological approaches employed by the authors to test the hypothesis are sound. The formulated scientific question is of great interest to the neuroscientist community.

This is an excellent paper that deserves to be published. However, some aspects of the current version of the manuscript should be clarified and improved. Please find them listed hereafter.

We thank the Reviewer for the positive feedback and comments and hope our clarifications have further improved the clarity of the manuscript.

R3.Q1. *It would be helpful to provide the sample sizes for each disorder in the main manuscript, so that the reader does not need to go to supplementary materials for this information.*

Thanks for this suggestion. We included the sample size for each disorder in the methods section of the revised manuscript, p. 23.

“Included mental disorders comprised ADHD ($n_{cases} = 733$, $n_{controls} = 539$), ASD ($n_{cases} = 1571$, $n_{controls} = 1651$), BD (type I and II, cumulated) ($n_{cases} = 1837$, $n_{controls} = 2582$), MDD ($n_{cases} = 1911$, $n_{controls} = 7663$), OCD ($n_{cases} = 1498$, $n_{controls} = 1436$), and SCZ (including schizophrenia spectrum diagnoses) ($n_{cases} = 4474$, $n_{controls} = 5098$).“

R3.Q2. *In Figure 1a, it is difficult to ascertain which regions have positive d values due to the grey colour used. Would it be possible to use a colour that is easier to visualise? In addition, although correlation matrices are provided for the HCP data, it would be helpful to see a visualisation of network hubs projected onto brain images.*

We changed the colormap for **Fig. 1A** and added visualizations of the HCP hubs in **Fig. 1B**. Please find the updated figure panels A and B below.

Figure 1. *Hubs and epicenters shaping transdiagnostic co-alteration patterns.* **A)** Condition-specific Cohen's *d* maps indicating case-control differences in cortical thickness. **B)** Normative resting-state (rs-fMRI) and diffusion-weighted connectivity matrices from the Human Connectome Project (HCP) and hubs (degree centrality). [...] ADHD = Attention-deficit/hyperactivity disorder; ASD = Autism spectrum disorder; BD = Bipolar disorder; MDD = Major depressive disorder; OCD = Obsessive-compulsive disorder; SCZ = Schizophrenia.

R3.Q3. *Could the authors please provide a justification for the inclusion of IQ as a covariate in their neuroimaging analyses for the ASD group. Similarly, was data for all bipolar, ASD, and ADHD participants collected at the same site? If not, is there a reason that site wasn't included a covariate for these samples?*

We are happy to further clarify. For the ASD cohort, **IQ** information was available for a subset of around 75% of the subjects included in the main analyses. For these subjects, previous work reported a mean group difference in IQ between patients and controls (mean controls = 111 SD=19.04 mean patients=103 sd=20.02; p-value <0.001), see table 1 of (van Rooij et al., 2018). Given the known effects of lower IQ on brain structure as well as alterations of this effect/association in ASD (see e.g. Misaki et al., 2012), the ENIGMA working group included this variable in their statistical models to correct for the confound of lower IQ for ASD effects. In the main publication of van Rooij et al. (2018), which presents the data we also used here, they included IQ as a fixed factor in all analyses. (Notably, a follow up publication comparing the ASD, ADHD and OCD groups, Boedhoe et al., (2020), addressed this issue further and reported similar patterns of ASD effects with and without correction for IQ.) We have now further clarified this in the Methods, p. 23.

“As previous studies have shown associations between IQ and brain structure as well as alterations of this association in ASD, IQ was included as a covariate in the ASD sample.”

Regarding the site correction in ASD, ADHD, and BD samples, data was indeed site corrected. However, in these meta-analyses, site was not included as a covariate but as a random effect in the mixed-effect model. We agree that including only covariate descriptions in the sample description table is misleading and included this information in the supplementary **Table S2**.

Table S2. Sample Demographics.

Disorder	site s	Weighted mean (cases)	age	Weighted mean (controls)	age	n	Covariates
Schizophrenia spectrum	39	32.3 ^a		34.5 ^a		Cases: 4474 Controls: 5098 Total: 9572	age, sex, scan site
Attention deficit hyperactivity disorder	36		32.97			Cases: 733 Controls: 539 Total: 1272	age, sex, scan site*
Autism spectrum disorder	49	15.4		15.8		Cases: 1571 Controls: 1651 Total: 3222	age, sex, IQ, scan site*
Bipolar disorder	28	38.4 ^a		35.6 ^a		Cases: 1837 Controls: 2582 Total: 4419	age, sex, scan site*
Major depressive disorder	20	44.8 ^a		54.6 ^a		Cases: 1911 Controls: 7663 Total: 9574	age, sex, scan site
Obsessive-compulsive disorder	27	32.1		30.5		Cases: 1498 Controls: 1436 Total: 2934	age, sex, scan site

Adapted from Radonjić et al. . ^a = weighted mean computed by Radonjić et al. . * = In this study, site was included as a random effect in a mixed-effect model and not as a covariate. IQ = Intelligence quotient.

R3.Q4. Can the authors confirm that for the schizophrenia sample, all participants had a diagnosis of schizophrenia and not other schizophrenia-spectrum disorders? Previous studies have shown neuroimaging differences between schizophrenia-spectrum diagnoses (e.g. schizoaffective, schizophrenia, psychosis NOS, etc.), which may be important to consider in the current study.

We thank the Reviewer for noting this. The original ENIGMA study in which schizophrenia case-control summary statistics were published (van Erp et al., 2018) included primarily, but not only, schizophrenia patients. Indeed, some of the participating sites did include schizophrenia spectrum disorders, which is why we changed the sample description correspondingly (i.e., using the term “schizophrenia spectrum disorders”). We acknowledge the difficulties that come with within-disorder heterogeneity in general, and here among schizophrenia spectrum diagnoses. However, our study focuses on transdiagnostic phenomena and thus patterns that are shared between disorders rather than what sub-clusters emerge within and between diagnoses. Schizophrenia itself, without related spectrum disorders, is already associated with rather strong patient heterogeneity (which, however, also applies to other psychiatric diagnoses such as MDD). The inclusion of spectrum diagnoses should have, in theory, provided us with a summary statistics map reflecting what boils down to be shared between heterogeneous schizophrenia (spectrum) patients. As we combine these disorder maps to further identify what the underlying shared features between diagnostic categories are, we believe that it is indeed beneficial when the disorder-specific maps themselves reflect the overlap between patients with heterogeneous symptomatology within a diagnostic category. Such a map would then provide a generalizable pattern, present across subclusters. At the same time, we agree that investigating both

disorder-specific patterns and also heterogeneity between highly related (spectrum-)diagnoses form a crucial complementary line of research that we hope to be able to link to our transdiagnostic findings in the future. We clarified on the inclusion of schizophrenia spectrum disorders in the introduction, p.4

“In this study, we identified hubs of transdiagnostic co-alteration networks and disease epicenters using meta-analytical maps for six mental disorders (autism spectrum disorder (ASD), attention-deficit/hyperactivity disorder (ADHD), major depressive disorder (MDD), schizophrenia spectrum disorders (SCZ), bipolar disorder (BD), and obsessive-compulsive disorder (OCD)), provided by the ENIGMA consortium.”

Methods, p. 23

“Included mental disorders comprised ADHD (ncases = 733, ncontrols = 539), ASD (ncases = 1571, ncontrols = 1651), BD (type I and II, cumulated) (ncases = 1837, ncontrols = 2582), MDD (ncases = 1911, ncontrols = 7663), OCD (ncases = 1498, ncontrols = 1436), and SCZ (including schizophrenia spectrum diagnoses) (ncases = 4474, ncontrols = 5098).”

and in the limitations/future implications section of the Discussion, p. 20.

“While our findings underline the relevance of transdiagnostic approaches, they do not contradict the existence of etiological and phenomenological differences between psychiatric diagnoses. Our transdiagnostic approach does not capture heterogeneity within and between highly related diagnostic categories, as expected to be present e.g. within included SCZ (schizophrenia spectrum disorders) and BD (type I and II combined) samples. However, shared features crossing diagnostic boundaries are likely also an important factor contributing to within-disorder heterogeneity. Moreover, individuals may be diagnosed with multiple different disorders across their lifespan. Understanding which neurobiological principles drive a shared and differential basis underlying the spectrum of varying mental disorders is a crucial piece of the puzzle of the biological origin of disorder variability. Yet, investigating both disorder-specific phenomena and heterogeneity within (spectrum-)diagnoses forms a crucial line of research that will continue to complement our transdiagnostic findings in the future. Presented cortex-wide co-alteration features shall facilitate and provide a new transdiagnostic coordinate frame for such insights.”

R3.Q5. What is missing for me is an examination of within-diagnosis covariance to supplement the transdiagnostic findings. Do the findings presented in Figure 1 hold across diagnostic groups, how much overlap is there between diagnoses? It is certainly interesting to see the transdiagnostic findings, however, I am left wondering what the similarities and differences might be across different diagnoses. Are cortical alterations driven by hubs of prominent covariance and epicentres in every disorder? Are findings the same for regions with reduced cortical thickness and those with increased cortical thickness?

We thank the Reviewer for these interesting points. To examine within-diagnosis covariance, we studied cross-cortical similarities of illness effects via absolute differences in Cohen’s d values of the disorder-specific summary scores between regions (Huntenburg et al., 2018; Sha et al., 2022). For each disorder, we investigated the correlation between a parcel’s whole-brain *transdiagnostic* covariance profile and a parcel’s whole-brain *disorder-specific* covariance profile (see **Fig. 3** below). This approach revealed the topography of the varying degrees to which within-disorder covariance aligned with transdiagnostic

covariance profiles. Mirroring other transdiagnostic findings presented in the manuscript, most disorders show highest similarity to shared patterns in heteromodal cortices. At the same time, the different disorders all have unique associations with the transdiagnostic pattern.

To address the question whether reported hubs and epicenters may promote cortical alterations in every disorder, we linked transdiagnostic hubs and epicenters to disorder-specific information. While transdiagnostic hubs correlated with illness effect maps in SCZ ($r = 0.76$, $p_{spin} < .0001$), BD ($r = 0.66$, $p_{spin} = 0.001$), and OCD ($r = 0.26$, $p_{spin} = 0.03$), this was not the case for ASD ($r = 0.07$, $p_{spin} = 0.31$) and MDD ($r = -0.05$, $p_{spin} = 0.41$), and ADHD showed a negative correlation ($r = -0.42$, $p_{spin} = 0.003$). Similarly, disorder-specific epicenters overlapped with transdiagnostic epicenters in SCZ, BD, and OCD, and in part in MDD and ASD (see **Fig 3E**). ADHD showed no significant epicenters. At the same time, we found that disorders sharing patterns of illness effects are positioned more closely together in a transdiagnostic covariance space framed by G1 and G2 (**Fig. 3F**). Thus, in line with our interpretation, it seems that the transdiagnostic covariance framework may describe shared biological axes within which individual disorders vary.

Further investigating *similarities and differences* across disorders, we observed that regions most strongly affected across disorders were primarily found in heteromodal cortices (especially medial temporal and lingual gyri; **Supplementary Fig. S1B**). Conversely, unimodal regions showed consistently low impact. Importantly, transdiagnostic co-alteration hubs presented in the manuscript were positively correlated with most consistently affected regions ($r = 0.42$, $p_{spin} < .0001$). This suggests that co-alteration hubs may to a larger extent capture shared illness effects than shared retainment, and regions consistently affected across disorders are also more likely embedded as hubs within co-alteration networks.

Regarding differences in regions showing *cortical thickness increases vs decreases*, we observed that transdiagnostic co-alteration hubs were mainly found in regions showing strongest thickness reductions across disorders ($r = 0.334$; $p_{spin} = 0.01$), and were anti-correlated with regions showing thickness increases ($r = -0.30$; $p_{spin} = 0.02$; see **Supplementary Fig. S1C&D** below). This indicates that the observed effect may be largely driven by shared patterns of thinning rather than relative thickness increases. This is likely related to the fact that thickness reductions are more abundant than increases across disorders (see **Fig. 1A** of Main manuscript) and thus are more likely captured by transdiagnostic features. The cortical topography of cortical thinning during maturation has been linked to expression patterns of genes involved in dendritic architecture (mainly spines and dendrites) and myelin, and associated with multiple psychiatric disorders. That is, there seems to be a molecular association between dysregulation of cortical cytoarchitecture during development and pathophysiological pathways of mental illness (Parker et al., 2020).

We have now added these analyses in the Main manuscript, Results, p.5.

*“When studying which regions are most strongly and consistently affected across disorders via the sum of normalized illness effect maps (see **Supplementary Fig. S1B**), we observed a significant correlation with transdiagnostic co-alteration hubs ($r = 0.42$, $p_{spin} < .0001$), suggesting hubs are placed in regions with shared impact. This effect predominated for shared thickness reductions ($r = 0.334$; $p_{spin} = 0.01$) rather than relative increases ($r = -0.30$; $p_{spin} = 0.02$.”*

[...] p. 12

“Embedding of individual disorders within a transdiagnostic co-alteration space

*Having established several features guiding a transdiagnostic co-alteration network, we last aimed to investigate the positioning of individual disorders within this continuous transdiagnostic space. To this end, we first investigated the correspondence between a parcel’s whole-brain transdiagnostic covariance profile and a parcel’s whole-brain disorder-specific covariance profile (see **Fig. 3A&B** below). While associations with the transdiagnostic pattern vary between disorders and across the cortex, most disorders showed highest similarity to shared patterns in heteromodal cortices. This mirrors other findings presented here which suggest heteromodal cortices as regions that not only tend to be affected, but also tend to be affected similarly across disorders and in a synchronized manner across the cortex. Next, we compared the degree of similarity between disorders and their embedding within the transdiagnostic co-alteration space. Replicating what previous transdiagnostic studies have shown (Radonjic et al., 2019; Opel et al., 2020), we observed a cluster composed of SCZ, BD, and OCD, while ADHD and ASD stayed separate (**Fig. 3 C&D**). In contrast to clustering approaches, our cross-disorder covariance approach aimed to describe a transdiagnostic organizational space in which disorder effects occur. Indeed, we find that disorders that cluster together, such as SZC, BD, and OCD show a similar placement within this transdiagnostic co-alteration framework (see **Fig. 3D-F**). While transdiagnostic hubs correlated with illness effect maps in SCZ ($r = 0.76$, $p_{spin} < .0001$), BD ($r = 0.66$, $p_{spin} = 0.001$), and OCD ($r = 0.26$, $p_{spin} = 0.03$), this was not the case for ASD ($r = 0.07$, $p_{spin} = 0.31$) and MDD ($r = -0.05$, $p_{spin} = 0.41$), and ADHD showed a negative correlation ($r = -0.42$, $p_{spin} = 0.003$). Similarly, disorder-specific epicenters overlapped with transdiagnostic epicenters in SCZ, BD, and OCD, and in part in MDD (see **Fig. 3E**), whereas ADHD and ASD showed no significant disorder-specific epicenters in the first place. Together, illness effect maps relate to similar degrees to transdiagnostic co-alteration hubs, are linked to epicenters that overlap to similar degrees with transdiagnostic epicenters (**Fig. 3E**), and are positioned more closely together in a transdiagnostic covariance space framed by G1 and G2 (**Fig. 3F**). However, we also observe that disorders which show some similarity but are allocated to different clusters, such as MDD and ADHD, are positioned closer to each other in our continuous transdiagnostic space, crossing cluster boundaries. Overall, disorders that show similar profiles of thickness impact are placed at similar positions within a transdiagnostic coordinate space.”*

Discussion, p. 19.

“Last, we aimed to investigate how the proposed transdiagnostic co-alteration space, made framed by both transdiagnostic covariance gradients, compares to previous descriptions of cross-disorder similarities and disorder clusters. That is, our cross-disorder covariance approach generates a continuous space within which disorders vary with respect to their topography of similarity to transdiagnostic patterns across the cortex. While we indeed find that positions of disorders within this space converge with their allocation to disorder clusters, the co-alteration framework captures both similarities within and between clusters in a continuous manner. By embedding illness effects within a space shaped by genetic and maturational processes, we gain further insight in differentiable neurobiological mechanisms underlying individual disorders. Indeed, the first gradient, stretching between frontal and temporal regions showed similarities with a previously described anterior-posterior axis along the cortical mantle (Valk, 2020). Previous work has indicated differentiable spatial patterns of co-maturation and development along multiple spatial axes, indicating the interplay of multiple neurodevelopmental mechanisms across the cortex (Valk, 2020; Zhu, 2018; Fornito, 2019). The observed systematic alterations along such axes across disorders may reflect differential disruptions in pre- and post-natal neurodevelopment. Future work may evaluate potential

causes and critical time windows of development within this framework, enhancing our understanding of the ontogeny of cortical organization in health and disorder. Moreover, we observed that, for most disorders, convergence between disorder-specific and transdiagnostic covariance is highest in heteromodal cortices, possibly linked to their placement within these neurodevelopmental axes (Baum et al., 2021; Paquola et al., 2019), supporting these regions as targets of transdiagnostic investigations. Future work may evaluate potential causes and critical time windows of development within this framework, enhancing our understanding of the ontogeny of cortical organization in health and disorder.”

[...] p. 20

“It is of note that, although disorder impact generally converged in heteromodal regions and linked to transdiagnostic covariance gradients, each disorder showed a unique embedding within our framework. For example, though we observed widespread coupling between transdiagnostic and disorder specific covariance networks in ASD and ADHD, and marked association with the principal transdiagnostic covariance gradient, there was only reduced correspondence with the epicenter framework, indicating disrupted relationship between disorder hubs and connectivity profiles. Conversely, MDD showed in particular correspondence with transdiagnostic patterns in ventral PFC, subgenual anterior cingulate, somatosensory cortex and nucleus accumbens, but showed reduced correspondence with transdiagnostic epicenters and the principal transdiagnostic gradients. It is possible that MDD, being at the center of the 2D gradient space and showing highest similarity with both ADHD and OCD, can be best described by yet another axis not captured in the current framework which is dominated by neurodevelopmental patterning. The future work expanding our framework to more disorders as well as atlases with higher granularity may be able to further pin-point differential axes of embedding for different disorders.”

Methods, p. 28.

“Association between disorder-specific illness effect patterns with transdiagnostic findings

Last, we aimed to understand the degree to which cortical alterations observed in individual disorders are reflected in described transdiagnostic features. To this end, we first examined cross-cortical similarities of illness effects within disorders (Huntenburg et al., 2017; Sha et al., 2022), via absolute differences in Cohen’s d values between regions. We then correlated each parcel’s disorder-specific whole-brain covariance profile with the previously described transdiagnostic covariance profile. This allowed us to investigate disorder-specific cortical topographies of varying regional associations with transdiagnostic patterns. Second, we examined the similarity of illness effect maps among disorders via pair-wise correlations and applied hierarchical clustering to the resulting cross-disorder correlation matrix. These steps allowed us to investigate how disorders with varying similarity to each other and to transdiagnostic features described in this study are positioned in the proposed transdiagnostic covariance space. To this end, we correlated the transdiagnostic co-alteration hub map with disorder-specific Cohen’s d maps, and computed disorder-specific epicenters by systematically correlating each region’s normative connectivity profile (rs-fMRI and DTI) to disorder-specific Cohen’s d maps. We then assessed the overlap between disorder-specific and transdiagnostic epicenters in percent, and combined this with the association to transdiagnostic hubs in a 2D space. Similarly, we examined the correlation between transdiagnostic gradients and disorder-specific Cohen’s d maps in a 2D space framed by G1 and G2. Together, these analyses revealed

how individual disorders are embedded in relation to each other within a transdiagnostic coordinate frame.”

and visualized these additional findings in Figure 3:

Figure 3. Embedding of six disorders within transdiagnostic co-alteration networks. A) Computation of transdiagnostic and within-disorder co-alteration matrices. B) Region-wise correspondence between disorder-specific and transdiagnostic co-alteration profiles. Disorder-specific inter-regional difference scores were inverted so that higher correlations with transdiagnostic patterns indicate higher coupling. C) Similarity of illness effects between disorders, i.e., correlations of Cohen's d maps, and how they cluster together (D). Position of individual disorders within a transdiagnostic co-alteration space based on E) the correlation between transdiagnostic hubs and Cohen's d maps (x-axis) and the overlap between transdiagnostic and disorder-specific epicenters (y-axis); and F) the correlation between transdiagnostic gradients G1 and G2 and Cohen's d maps.

and Supplementary Figure S7.

Figure S7. *Overlaps in disorder-specific disease epicenters.* In order to examine to which degree transdiagnostic epicenters also reflect individual disorder's epicenters, we quantified the overlap of functional (A) and structural (B) epicenters. Epicenter maps for each disorder were binarized, labeling a region as epicenter or no epicenter, and then summed, reflecting in how many disorders a region forms an epicenter. Epicenter maps for individual disorders are depicted in C).

R3.Q6. *Is it possible that the number of subjects across diagnoses could bias findings? I.e. could diagnostic groups with larger samples bias findings towards those groups? Is it possible to perform supplementary analyses with the number of subjects per group held constant at the same size to determine how sample size might impact results?*

We thank the Reviewer for the comment. As our study is based on publicly available summary statistics, it is unfortunately not possible to repeat the current analyses on sub-samples with constant n . As written above in our response to #R2.Q4, Cohen's d values theoretically already account for sample sizes, and results were not influenced when including sample size as a covariate: correlation of n -corrected co-alteration hub map with original co-alteration hub map: $r = 0.89$, $p < .0001$; Gradient 1: $r = 0.9861$, $p < .0001$; Gradient 2: $r = 0.9577$, $p < .0001$.

We included these additional quality checks in the supplementary sections “*Robustness of cross-disorder co-alteration network hubs*” and “*Robustness of transdiagnostic gradients*”, as well as in **Figure S2**.

“*Correcting the cross-disorder co-alteration matrix for sample sizes via a partial correlation did not impact resulting hubs (correlation of n-corrected co-alteration hub map with original co-alteration hub map: $r = 0.89$, $p_{spin} < .0001$; Fig. S2A).*”

[...]

“*Fourth, gradients were not impacted by correcting for sample sizes of underlying disorder samples as tested by including sample sizes as covariates in the computation of the cross-disorder co-alteration matrix via partial correlation (G1: $r = 0.99$, p_{spin} ; G2: $r = 0.96$, p_{spin} ; Fig. S2B).*”

Figure S2. *Robustness of co-alteration hubs and transdiagnostic gradients to parameter manipulations.* Values indicate correlation with original hubs/gradients after parameter manipulation. **A)** Co-alteration hubs based on co-alteration matrix with different cut-offs or corrected for sample-size (n-corrected) per disorder. **B)** Left: Corrected for average illness effects and sample size, or Laplacian eigenmap or principal component analysis (PCA) as dimensionality reduction techniques, or co-alterations based on spearman's rho. Middle: Co-alteration matrix cut-offs. Right: Constructing gradients based on five disorders only, highlighting the contribution of single disorders. *G1 and G2 are switched for autism spectrum disorder (ASD). BD = Bipolar disorder, ADHD = Attention-deficit/hyperactivity disorder, MDD = Major depressive disorder, OCD = Obsessive compulsive disorder, SCZ = Schizophrenia spectrum disorder.

References

- Alexander-Bloch, A. F., Mathias, S. R., Fox, P. T., Olvera, R. L., Göring, H. H. H., Duggirala, R., Curran, J. E., Blangero, J., & Glahn, D. C. (2019). Human Cortical Thickness Organized into Genetically-determined Communities across Spatial Resolutions. *Cerebral Cortex*, *29*(1), 106–118. <https://doi.org/10.1093/cercor/bhx309>
- Alexander-Bloch, A. F., Shou, H., Liu, S., Satterthwaite, T. D., Glahn, D. C., Shinohara, R. T., Vandekar, S. N., & Raznahan, A. (2018). On testing for spatial correspondence between maps of human brain structure and function. *Neuroimage*, *178*, 540–551. <https://doi.org/10.1016/j.neuroimage.2018.05.070>
- Arnatkeviciute, A., Markello, R. D., Fulcher, B., Masic, B., & Fornito, A. (2022). *Towards best practices for imaging transcriptomics*. OSF Preprints. <https://doi.org/10.31219/osf.io/y8ftn>
- Baum, G. L., Flournoy, J. C., Glasser, M. F., Harms, M. P., Mair, P., Sanders, A., Barch, D. M., Buckner, R. L., Bookheimer, S., Dapretto, M., Smith, S., Thomas, K. M., Yacoub, E., Essen, D. C. V., & Somerville, L. H. (2021). *Graded Variation In Cortical T1w/T2w Myelination During Adolescence* (p. 2021.12.06.471432). bioRxiv. <https://doi.org/10.1101/2021.12.06.471432>
- Bernhardt, B. C., Smallwood, J., Keilholz, S., & Margulies, D. S. (2022). Gradients in brain organization. *NeuroImage*, *251*, 118987. <https://doi.org/10.1016/j.neuroimage.2022.118987>
- Boedhoe, P. S. W., van Rooij, D., Hoogman, M., Twisk, J. W. R., Schmaal, L., Abe, Y., Alonso, P., Ameis, S. H., Anikin, A., Anticevic, A., Arango, C., Arnold, P. D., Asherson, P., Assogna, F., Auzias, G., Banaschewski, T., Baranov, A., Batistuzzo, M. C., Baumeister, S., ... van den Heuvel, O. A. (2020). Subcortical Brain Volume, Regional Cortical Thickness, and Cortical Surface Area Across Disorders: Findings From the ENIGMA ADHD, ASD, and OCD Working Groups. *The American Journal of Psychiatry*, *177*(9), 834–843. <https://doi.org/10.1176/appi.ajp.2020.19030331>
- Dell’Osso, L., Lorenzi, P., & Carpita, B. (2019). The neurodevelopmental continuum towards a neurodevelopmental gradient hypothesis. *Journal of Psychopathology*, *25*, 179–182.
- Fornito, A., Arnatkevičiūtė, A., & Fulcher, B. D. (2019). Bridging the gap between connectome and transcriptome. *Trends in Cognitive Sciences*, *23*(1), 34–50.
- Fulcher, B. D., Arnatkeviciute, A., & Fornito, A. (2021). Overcoming false-positive gene-category enrichment in the analysis of spatially resolved transcriptomic brain atlas data. *Nature Communications*, *12*(1), 2669. <https://doi.org/10.1038/s41467-021-22862-1>
- Huntenburg, J. M., Bazin, P. L., & Margulies, D. S. (2018). Large-Scale Gradients in Human Cortical Organization. *Trends Cogn Sci*, *22*(1), 21–31. <https://doi.org/10.1016/j.tics.2017.11.002>
- Huntenburg, J. M., Bazin, P.-L., Goulas, A., Tardif, C. L., Villringer, A., & Margulies, D. S. (2017). A Systematic Relationship Between Functional Connectivity and Intracortical Myelin in the Human Cerebral Cortex. *Cerebral Cortex (New York, N.Y.: 1991)*, *27*(2), 981–997. <https://doi.org/10.1093/cercor/bhx030>
- Insel, T., Cuthbert, B., Garvey, M., Heinssen, R., Pine, D. S., Quinn, K., Sanislow, C., & Wang, P. (2010). Research Domain Criteria (RDoC): Toward a New Classification Framework for Research on Mental Disorders. *American Journal of Psychiatry*, *167*(7), 748–751. <https://doi.org/10.1176/appi.ajp.2010.09091379>
- Katzman, M. A., Bilkey, T. S., Chokka, P. R., Fallu, A., & Klassen, L. J. (2017). Adult ADHD and comorbid disorders: Clinical implications of a dimensional approach. *BMC Psychiatry*, *17*(1), 302. <https://doi.org/10.1186/s12888-017-1463-3>
- Larivière, S., Rodríguez-Cruces, R., Royer, J., Caligiuri, M. E., Gambardella, A., Concha, L., Keller, S. S., Cendes, F., Yasuda, C., Bonilha, L., Gleichgerrcht, E., Focke, N. K., Domin, M., Von Podewills, F., Langner, S., Rummel, C., Wiest, R., Martin, P., Kotikalapudi, R., ... Bernhardt, B. C. (2020). Network-based atrophy modeling in the common epilepsies: A worldwide ENIGMA study. *Science Advances*, *6*(47), eabc6457. <https://doi.org/10.1126/sciadv.abc6457>
- Lee, P. H., Anttila, V., Won, H., Feng, Y.-C. A., Rosenthal, J., Zhu, Z., Tucker-Drob, E. M., Nivard, M. G., Grotzinger, A. D., Posthuma, D., Wang, M. M.-J., Yu, D., Stahl, E. A., Walters, R. K., Anney, R. J. L., Duncan, L. E., Ge, T., Adolfsson, R., Banaschewski, T., ... Smoller, J. W. (2019). Genomic Relationships, Novel Loci, and Pleiotropic Mechanisms across Eight

- Psychiatric Disorders. *Cell*, 179(7), 1469-1482.e11. <https://doi.org/10.1016/j.cell.2019.11.020>
- Margulies, D. S., Ghosh, S. S., Goulas, A., Falkiewicz, M., Huntenburg, J. M., Langs, G., Bezgin, G., Eickhoff, S. B., Castellanos, F. X., Petrides, M., Jefferies, E., & Smallwood, J. (2016). Situating the default-mode network along a principal gradient of macroscale cortical organization. *Proceedings of the National Academy of Sciences*, 113(44), 12574–12579. <https://doi.org/10.1073/pnas.1608282113>
- Marshall, M. (2020). The hidden links between mental disorders. *Nature*, 581(7806), 19–21. <https://doi.org/10.1038/d41586-020-00922-8>
- Menon, V. (2011). Large-scale brain networks and psychopathology: A unifying triple network model. *Trends in Cognitive Sciences*, 15(10), 483–506.
- Misaki, M., Wallace, G. L., Dankner, N., Martin, A., & Bandettini, P. A. (2012). Characteristic cortical thickness patterns in adolescents with autism spectrum disorders: Interactions with age and intellectual ability revealed by canonical correlation analysis. *NeuroImage*, 60(3), 1890–1901. <https://doi.org/10.1016/j.neuroimage.2012.01.120>
- Opel, N., Goltermann, J., Hermesdorf, M., Berger, K., Baune, B. T., & Dannlowski, U. (2020). Cross-Disorder Analysis of Brain Structural Abnormalities in Six Major Psychiatric Disorders: A Secondary Analysis of Mega- and Meta-analytical Findings From the ENIGMA Consortium. *Biological Psychiatry*, 88(9), 678–686. <https://doi.org/10.1016/j.biopsych.2020.04.027>
- Paquola, C., Bethlehem, R. A., Seidlitz, J., Wagstyl, K., Romero-Garcia, R., Whitaker, K. J., Vos De Wael, R., Williams, G. B., Vértes, P. E., Margulies, D. S., Bernhardt, B., & Bullmore, E. T. (2019). Shifts in myeloarchitecture characterise adolescent development of cortical gradients. *ELife*, 8. <https://doi.org/10.7554/elife.50482>
- Park, B., Kebets, V., Larivière, S., Hettwer, M. D., Paquola, C., Rooij, D. van, Buitelaar, J., Franke, B., Hoogman, M., Schmaal, L., Veltman, D. J., Heuvel, O. van den, Stein, D. J., Andreassen, O. A., Ching, C. R. K., Turner, J., Erp, T. G. M. van, Evans, A. C., Dagher, A., ... Bernhardt, B. C. (2021). *Multilevel neural gradients reflect transdiagnostic effects of major psychiatric conditions on cortical morphology* (p. 2021.10.29.466434). <https://doi.org/10.1101/2021.10.29.466434>
- Parker, N., Patel, Y., Jackowski, A. P., Pan, P. M., Salum, G. A., Pausova, Z., Paus, T., & Saguenay Youth Study and the IMAGEN Consortium. (2020). Assessment of Neurobiological Mechanisms of Cortical Thinning During Childhood and Adolescence and Their Implications for Psychiatric Disorders. *JAMA Psychiatry*, 77(11), 1127–1136. <https://doi.org/10.1001/jamapsychiatry.2020.1495>
- Patel, Y., Parker, N., Shin, J., Howard, D., French, L., Thomopoulos, S. I., Pozzi, E., Abe, Y., Abé, C., Anticevic, A., Alda, M., Aleman, A., Alloza, C., Alonso-Lana, S., Ameis, S. H., Anagnostou, E., McIntosh, A. A., Arango, C., Arnold, P. D., ... Paus, T. (2021). Virtual Histology of Cortical Thickness and Shared Neurobiology in 6 Psychiatric Disorders. *JAMA Psychiatry*, 78(1), 47. <https://doi.org/10.1001/jamapsychiatry.2020.2694>
- Plana-Ripoll, O., Pedersen, C. B., Holtz, Y., Benros, M. E., Dalsgaard, S., De Jonge, P., Fan, C. C., Degenhardt, L., Ganna, A., & Greve, A. N. (2019). Exploring comorbidity within mental disorders among a Danish national population. *JAMA Psychiatry*, 76(3), 259–270. <https://doi.org/10.1001/jamapsychiatry.2018.3658>
- Radonjic, N. V., Hess, J. L., Rovira, P., Andreassen, O., Buitelaar, J. K., Ching, C. R. K., Franke, B., Hoogman, M., Jahanshad, N., McDonald, C., Schmaal, L., Sisodiya, S. M., Stein, D. J., van den Heuvel, O. A., van Erp, T. G. M., van Rooij, D., Veltman, D. J., Thompson, P., & Faraone, S. V. (2021). Structural brain imaging studies offer clues about the effects of the shared genetic etiology among neuropsychiatric disorders. *Mol Psychiatry*. <https://doi.org/10.1038/s41380-020-01002-z>
- Schmaal, L., Hibar, D. P., Samann, P. G., Hall, G. B., Baune, B. T., Jahanshad, N., Cheung, J. W., van Erp, T. G. M., Bos, D., Ikram, M. A., Vernooij, M. W., Niessen, W. J., Tiemeier, H., Hofman, A., Wittfeld, K., Grabe, H. J., Janowitz, D., Bulow, R., Selonke, M., ... Veltman, D. J. (2017). Cortical abnormalities in adults and adolescents with major depression based on brain scans from 20 cohorts worldwide in the ENIGMA Major Depressive Disorder Working Group. *Mol Psychiatry*, 22(6), 900–909. <https://doi.org/10.1038/mp.2016.60>
- Sha, Z., van Rooij, D., Anagnostou, E., Arango, C., Auzias, G., Behrmann, M., Bernhardt, B., Bolte,

- S., Busatto, G. F., Calderoni, S., Calvo, R., Daly, E., Deruelle, C., Duan, M., Duran, F. L. S., Durston, S., Ecker, C., Ehrlich, S., Fair, D., ... Francks, C. (2022). Subtly altered topological asymmetry of brain structural covariance networks in autism spectrum disorder across 43 datasets from the ENIGMA consortium. *Molecular Psychiatry*, 27(4), 2114–2125. <https://doi.org/10.1038/s41380-022-01452-7>
- Shafiei, G., Markello, R. D., Makowski, C., Talpalaru, A., Kirschner, M., Devenyi, G. A., Guma, E., Hagmann, P., Cashman, N. R., Lepage, M., Chakravarty, M. M., Dagher, A., & Mišić, B. (2020). Spatial Patterning of Tissue Volume Loss in Schizophrenia Reflects Brain Network Architecture. *Biological Psychiatry*, 87(8), 727–735. <https://doi.org/10.1016/j.biopsych.2019.09.031>
- Valk, S., Xu, T., Margulies, D., Masouleh, S. K., Paquola, C., Goulas, A., Kochunov, P., Smallwood, J., Yeo, B. T. T., Bernhardt, B. C., & Eickhoff, S. B. (2020). Shaping brain structure: Genetic and phylogenetic axes of macroscale organization of cortical thickness. *Science Advances*, 6(39). <https://doi.org/DOI: 10.1126/sciadv.abb3417>
- van Erp, T. G. M., Walton, E., Hibar, D. P., Schmaal, L., Jiang, W., Glahn, D. C., Pearlson, G. D., Yao, N., Fukunaga, M., Hashimoto, R., Okada, N., Yamamori, H., Bustillo, J. R., Clark, V. P., Agartz, I., Mueller, B. A., Cahn, W., de Zwarte, S. M. C., Hulshoff Pol, H. E., ... Turner, J. A. (2018). Cortical Brain Abnormalities in 4474 Individuals With Schizophrenia and 5098 Control Subjects via the Enhancing Neuro Imaging Genetics Through Meta Analysis (ENIGMA) Consortium. *Biol Psychiatry*, 84(9), 644–654. <https://doi.org/10.1016/j.biopsych.2018.04.023>
- van Rooij, D., Anagnostou, E., Arango, C., Auzias, G., Behrmann, M., Busatto, G. F., Calderoni, S., Daly, E., Deruelle, C., Di Martino, A., Dinstein, I., Duran, F. L. S., Durston, S., Ecker, C., Fair, D., Fedor, J., Fitzgerald, J., Freitag, C. M., Gallagher, L., ... Buitelaar, J. K. (2018). Cortical and Subcortical Brain Morphometry Differences Between Patients With Autism Spectrum Disorder and Healthy Individuals Across the Lifespan: Results From the ENIGMA ASD Working Group. *Am J Psychiatry*, 175(4), 359–369. <https://doi.org/10.1176/appi.ajp.2017.17010100>
- Wei, Y., de Lange, S. C., Pijnenburg, R., Scholtens, L. H., Ardesch, D. J., Watanabe, K., Posthuma, D., & van den Heuvel, M. P. (2022). Statistical testing in transcriptomic-neuroimaging studies: A how-to and evaluation of methods assessing spatial and gene specificity. *Human Brain Mapping*, 43(3), 885–901. <https://doi.org/10.1002/hbm.25711>
- Zhou, J., Gennatas, E. D., Kramer, J. H., Miller, B. L., & Seeley, W. W. (2012). Predicting regional neurodegeneration from the healthy brain functional connectome. *Neuron*, 73(6), 1216–1227.
- Zhu, Y., Sousa, A. M. M., Gao, T., Skarica, M., Li, M., Santpere, G., Esteller-Cucala, P., Juan, D., Ferrández-Peral, L., Gulden, F. O., Yang, M., Miller, D. J., Marques-Bonet, T., Imamura Kawasawa, Y., Zhao, H., & Sestan, N. (2018). Spatiotemporal transcriptomic divergence across human and macaque brain development. *Science*, 362(6420), eaat8077. <https://doi.org/10.1126/science.aat8077>

REVIEWER COMMENTS

Reviewer #1 (Remarks to the Author):

Thank you for the comprehensive revision of the manuscript. While I still believe that this work would fit better in a journal with more specialised audience, the authors did do a good job in clarifying the rationale for the study and the general importance of their findings. As far as I can tell, other concerns have also been successfully addressed and it is clear the manuscript improved in its quality and clarity. Therefore, may the editors consider the findings of sufficient relevance for their journal, I have no further concerns regarding publication of this manuscript in its current form.

Reviewer #2 (Remarks to the Author):

The authors have addressed all issues. No other questions.

Reviewer #3 (Remarks to the Author):

I thank the authors for so diligently responding to my comments, and am happy for the revised manuscript to be published.